# A tissue-specific self-interacting chromatin domain forms independently of enhancer-promoter interactions

Jill M. Brown [1], Nigel A. Roberts [1], Bryony Graham[1], Dominic Waithe [2], Christoffer Lagerholm[2], Jelena M. Telenius[1], Sara De Ornellas [1], A. Marieke Oudelaar[1], Caroline Scott [1], Izabela Szczerbal[1,3], Christian Babbs[1], Mira T. Kassouf[1], Jim R. Hughes [1], Douglas R. Higgs [1] & Veronica J. Buckle [1]

Self-interacting chromatin domains encompass genes and their *cis*-regulatory elements; however, the three-dimensional form a domain takes, whether this relies on enhancer–promoter interactions, and the processes necessary to mediate the formation and maintenance of such domains, remain unclear. To examine these questions, here we use a combination of high-resolution chromosome conformation capture, a non-denaturing form of fluorescence in situ hybridisation and super-resolution imaging to study a 70 kb domain encompassing the mouse α-globin regulatory locus. We show that this region forms an erythroid-specific, decompacted, self-interacting domain, delimited by frequently apposed CTCF/cohesin binding sites early in terminal erythroid differentiation, and does not require transcriptional elongation for maintenance of the domain structure. Formation of this domain does not rely on interactions between the α-globin genes and their major enhancers, suggesting a transcription-independent mechanism for establishment of the domain. However, absence of the major enhancers does alter internal domain interactions. Formation of a loop domain therefore appears to be a mechanistic process that occurs irrespective of the specific interactions within.

[1] MRC Molecular Haematology Unit, MRC Weatherall Institute of Molecular Medicine, Oxford University, Oxford OX3 9DS, UK. [2] Wolfson Imaging Centre Oxford, MRC Weatherall Institute of Molecular Medicine, Oxford OX3 9DS, UK. [3] Present address: Department of Genetics and Animal Breeding, Poznan University of Life Sciences, Wolynska 33, 60-637 Poznan, Poland. Correspondence and requests for materials should be addressed to V.J.B. (email: veronica.buckle@imm.ox.ac.uk)

The biological activity of our genomes is not determined by the linear DNA sequence alone; a major current goal in biology is to characterise the three-dimensional (3D) architecture of chromatin in the interphase nucleus, to establish how it changes during development and differentiation and to elucidate the functional implications of that organisation. Proximity ligation and deep sequencing techniques have identified megabase-scale topologically associating domains (TADs) throughout the genome[1,2]. Within that framework there are sub-TAD structures or domains that appear to undergo reorganisation between cellular states[3–5]. Chromatin loops are thought to occur within such domains, bringing gene promoters into proximity with their cognate cis-acting regulatory elements within the same domain[6–10]. TADs and sub-TADs can be anchored by CTCF/cohesin[5], and such boundary sites have been shown as necessary to prevent abnormal enhancer–promoter contacts[11–13]. Hence a sub-TAD or loop domain can define the region of influence of an enhancer, which represents a fundamental mechanism for specificity and selectivity in gene regulation. What remains unclear is the three-dimensional form of the domain that must arise to support optimal enhancer–promoter interactions and what elements are important to mediate this structure.

To establish chromatin architecture at sequential stages of differentiation and in the absence of key elements, we have utilised the well-characterised murine α-globin regulatory locus. This locus is contained within a 70 kb self-interacting domain in which we have previously identified all cis-acting elements, including promoters, enhancers and CTCF/cohesin-binding sites (Fig. 1)[14]. In mES cells the α-globin promoters and five enhancers are not bound by transcription factors and the genes are silent[15]. Further, we detect no evidence of a strong self-interacting domain in mES cells, whereas such a structure is clearly present in differentiating erythroblasts[16]. Yet the largely convergent boundary elements are occupied by CTCF and cohesin in both cell types[17], suggesting that CTCF/cohesin are not the primary mediators of this tissue-specific domain formation.

The chromosome conformation technique next-generation (NG) Capture-C has demonstrated that the domain of interaction at the murine α-globin locus consists of multiple sampling interactions[14,16]. The increased contact frequency throughout the entire domain could be interpreted as a spatially confined volume, although the relationship of interaction frequencies to chromatin organisation and compaction is unclear[18] since this type of data represents an averaged view of chromatin conformation.

Here in addition to NG Capture-C, we use a modified fluorescence in situ hybridisation (FISH) approach to gain insight, at the level of the single cell, into the 3D structure of a well-defined sub-TAD domain, how that alters during erythroid differentiation and what roles key sequences within the domain play in modulating its formation. We are able to visualise and measure the configuration of the active α-globin domain on a cell-by-cell basis. We show that interactions between the α-globin genes and their enhancers are not required at this locus to initiate formation of a decompacted self-interacting domain and that maintenance of such a domain does not require ongoing transcriptional elongation.

## Results

**Definition of erythroid cell populations**. To determine the relationship between activation of the α-globin gene cluster and formation of a self-interacting domain, we examined mouse embryonic stem cells where α-globin is silent together with foetal liver cells from E12.5 mice (MFL), cultured and harvested ex vivo at two stages of differentiation. After an initial expansion period, foetal liver-derived erythroid cells are depleted for Ter119-

expressing differentiated cells. Cells in this population (MFL 0 h) correspond broadly to early erythroblasts in which all cis-elements are bound by transcription factors but there is little detectable α-globin expression (Fig. 1a–d). After 30 h culture in differentiation medium (MFL 30 h), the α-globin genes transcribe at maximal levels with very strong RNA-FISH signals (Fig. 1a–d). Each population is likely to contain cells at varying stages of differentiation however the bulk of the two populations are clearly distinct by FACS, morphology, reverse transcription (RT)-qPCR and RNA-FISH (Fig. 1a–d).

**The domain of interaction exists early in differentiation**. NG Capture-C tracks of the α-globin locus at MFL 30 h from four viewpoints are shown together with locations of the FISH probes used (Fig. 1e). The α-globin domain of interaction is indistinguishable to that previously identified in erythroblasts derived from adult spleen (Supplementary Fig. 1)[16] and demonstrates increased contact frequencies throughout the entire domain detected from the viewpoints of either the Hba1/2 promoters or the multispecies conserved sequences (MCS)-R2 enhancer. Viewpoints from boundary CTCF-binding sites however demonstrate an avoidance of the active α-globin domain and instead show a general interaction plateau over a region on the opposite side of the domain as previously reported[17]. NG Capture-C analysis of chromatin interaction frequencies[16] at the α-globin locus in MFL 0 h and 30 h populations indicates that the α-globin self-interacting domain is present and apparently equivalent in both populations, as is the β-globin domain (Fig. 2a, b). Concordant with evidence in neural development[19], we conclude that the globin self-interacting domains, absent in mouse ES cells[16], are already formed at an early stage of erythroid differentiation, prior to the onset of robust α-globin and β-globin transcription.

**FISH analysis rationale**. To ensure that we preserved nuclear structure of the cells being analysed, we developed a method RASER-FISH (resolution after single-strand exonuclease resection) for hybridizing to chromosomal loci without denaturing the DNA (Methods section). For FISH probes, we used a BAC (COMP) precisely covering the α-globin self-interacting domain, and two BACs (F1 and F2) flanking this region (Fig. 1). We also designed smaller (~7 kb) plasmid probes (Ex, E, A, C and Cx) to analyse chromatin organisation in finer detail (Fig. 1). The Anchor 'A' probe was located at the distal extremity of the domain and is the nearest region of unique sequence adjacent to the α-globin genes. All measurements involving the plasmid probes were made relative to this position. 'Ex' defines the proximal edge of the domain and 'E' was sited at the two major enhancer elements MCS-R1 and MCS–R2. Two control probes ('C' and 'Cx') were positioned outside of the self-interacting domain, in a region showing little interaction by NG Capture-C with the α-globin genes or their enhancers. These control probes are equidistant in linear sequence to A as upstream probes E and Ex respectively. Prior to the FISH experiments, we performed NG Capture-C on MFL 30 h erythroblasts and mES cells using capture oligonucleotides corresponding to the central points of each plasmid probe, to determine the interactions they detect across the locus (Supplementary Fig. 2). FISH with this probe panel then allowed us to measure 3D distances and volumes occupied by the chromatin within and outside the α-globin domain in single cells.

**Flanking chromatin interacts and domain chromatin decompacts**. Using BACs COMP, F1 and F2, we measured inter-probe distances in mES cells and MFL 0 h and 30 h erythroblasts with three paired combinations (COMP-F1, COMP-F2 and F1–F2)

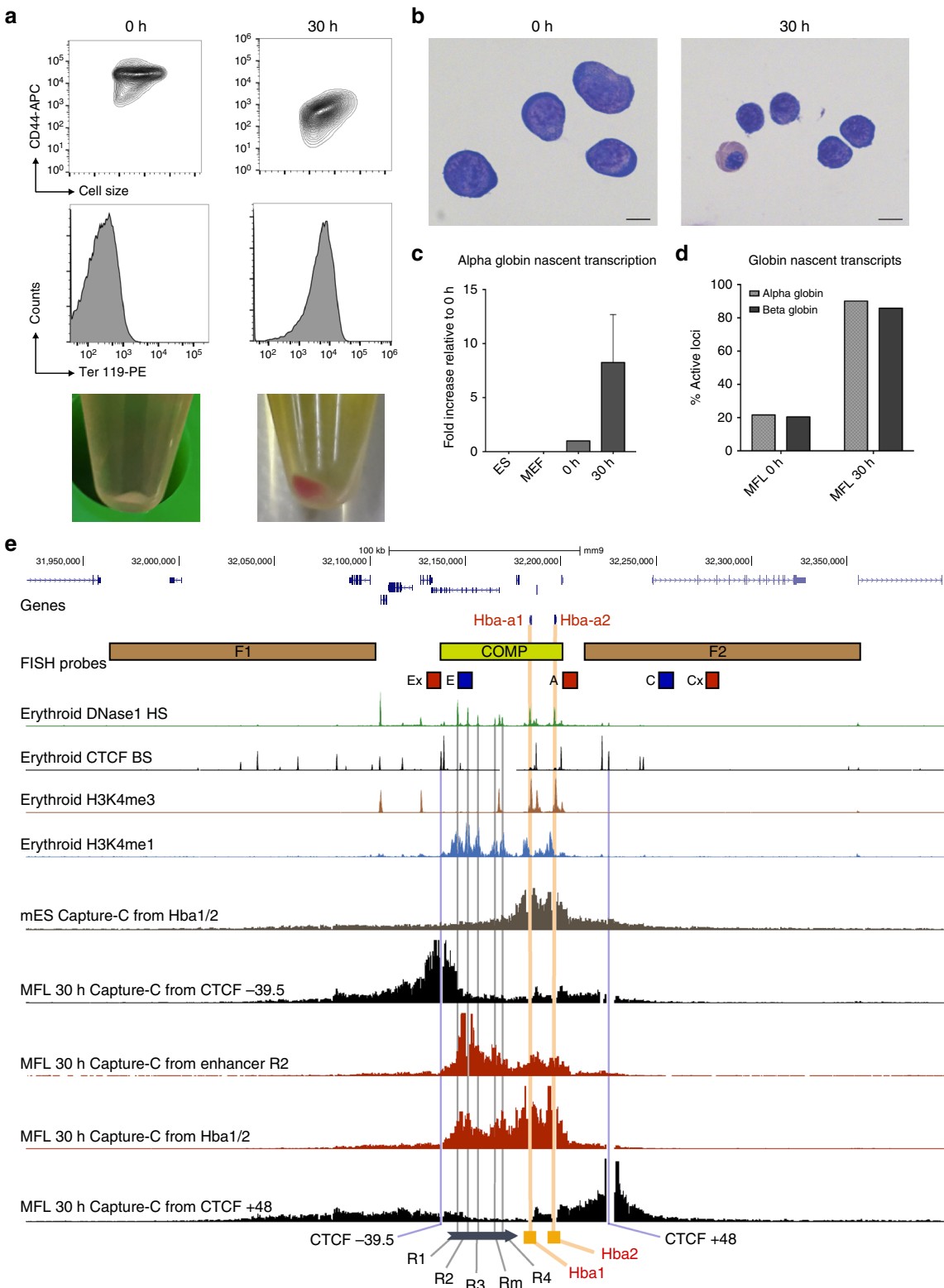

(Fig. 3a, b). We found no significant differences between COMP-F1 and COMP-F2 in mES cells, MFL 0 h or MFL 30 h erythroblasts (Fig. 3c, Supplementary Fig. 3). However, we noted that although the linear genomic distance between F1 and F2 is twice that of COMP-F1 or COMP-F2, the median inter-probe distances between F1 and F2 are nevertheless shorter at MFL 0 h (Fig. 3c). Importantly at MFL 30 h, we observed a further, highly significant shortening of F1–F2 measurements compared to COMP-F1 and COMP-F2 (Fig. 3c). By contrast, there is no significant difference between distance measurements for the three probe pairs in mES cells. To analyse this further, we calculated the Pearson's correlation for the BAC signal pairs to estimate the degree of signal overlap. There is marked overlap between F1 and F2 signals at 0 h (median coefficients 0.6–0.61) and further overlap at 30 h (median coefficients 0.66–0.75) in contrast to mES cells (median co-efficient 0.43). Thus it appears that the flanking regions of the

Fig. 1 Description of the mouse foetal liver-derived erythroblast populations and of the active murine α-globin locus. **a** Representative FACS plots of MFL erythroblasts defined by CD44/cell size and Ter119, at 0 h and after a further 30 h differentiation in vitro, identifying distinct populations at the two timepoints. Bottom images are representative cell pellets at the two timepoints demonstrating the presence of haemoglobin at MFL 30 h. **b** Representative cytospins of the MFL 0 h and 30 h cultures. Scale bar 5 μm. **c** Nascent *Hba* transcription relative to *18s* in the cell types and MFL timepoints indicated, from 3 biological replicates. Error bar is standard deviation. **d** RNA-FISH analysis of nascent transcription from the α-globin and β-globin genes at MFL 0 h and 30 h. *n* = 388 at MFL0h and 416 at MFL30h. **e** Map of the gene dense murine α-globin locus with *Hba* genes highlighted in red and positions of the FISH probes used in brown and lime green (BAC probes) and red and blue (plasmid probes). Gene browser tracks depict DNase1 hypersensitive sites (HS green), CTCF-binding sites (BS black), H3K4me3 (brown) and H3K4me1 (blue). NG Capture-C derived interaction frequencies are shown in mES cells from the *Hba1/2* viewpoints (grey), and in MFL 30 h from viewpoints (CTCF BS -39.5 (black), MCS-R1 (red), *Hba1/2* (red) and CTCF BS + 48 (black). The location of the *Hba* genes, the five murine enhancer elements and the CTCF BS −39.5 and +48 are marked against the browser tracks in yellow, grey and blue vertical bars, respectively

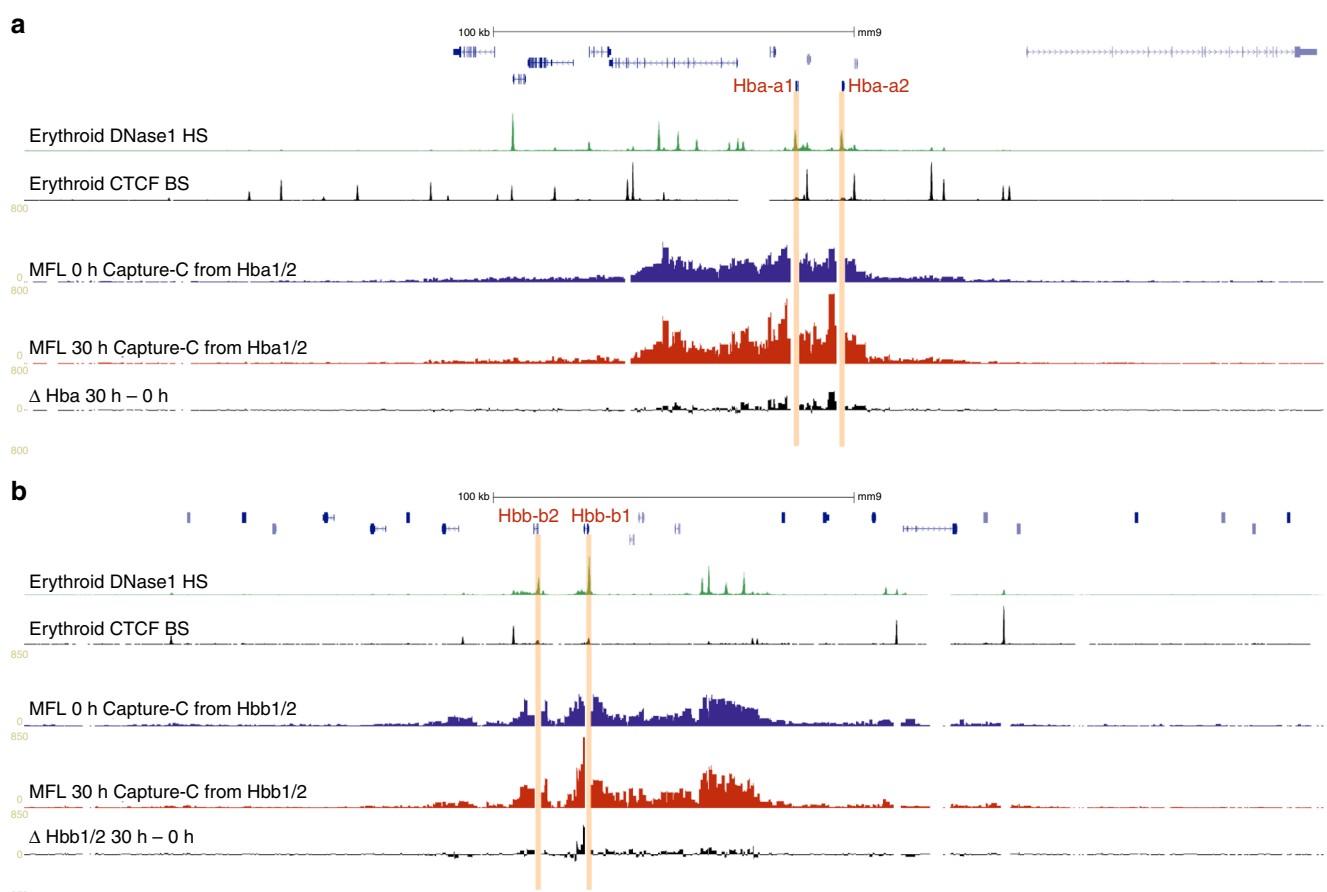

Fig. 2 Self-interacting domains are already formed at globin loci in MFL 0 h erythroblasts. **a** NG Capture-C tracks using *Hba1/2* as viewpoints in MFL erythroblasts at 0 h (blue) and 30 h (red) with a differential track (black) showing minimal changes between the two timepoints. **b** NG Capture-C tracks using *Hbb1/2* as viewpoints in MFL erythroblasts at 0 h and 30 h with a differential track showing minimal changes between the two timepoints

self-interacting domain are more frequently found in close proximity in erythroblasts compared to mES cells.

Next, to investigate the relationship between interaction frequency and chromatin compaction in the α-globin domain, we looked at the volume of the BAC probe signals as a relative measure of the degree of decompaction. The genomic length of the COMP probe is 64 kb compared to 139 kb for both F1 and F2 however the volume of the COMP BAC signal was greater at MFL 0 h than F1 or F2 and the difference was increased at MFL 30 h. By contrast in mES cells, the three sets of volume measurements were comparable (Fig. 3d). This indicates that chromatin within the domain is less compact relative to surrounding chromatin in erythroblasts so that rather than a compact structure, as might be

construed from interaction frequency data, this self-interacting domain is a decompacted, dynamically interacting region.

To further test these observations, we used the precisely positioned pairs of plasmid probes (A-E, A-C, A-Ex and A-Cx), to analyse intra-chromosomal distances in mES and erythroid cells (Fig. 4, Supplementary Fig. 4). As for the BAC probes, we found no significant differences in mES cells between measurements across the self-interacting domain versus the control region (A-Ex versus A-Cx). This finding is supported by NG Capture-C data from the viewpoints of all five plasmid probes where only immediate proximity interactions are detected (Supplementary Fig. 2), indicating that no domain of interaction is present at this site in mES cells. Analysing early erythroblasts (MFL 0 h),

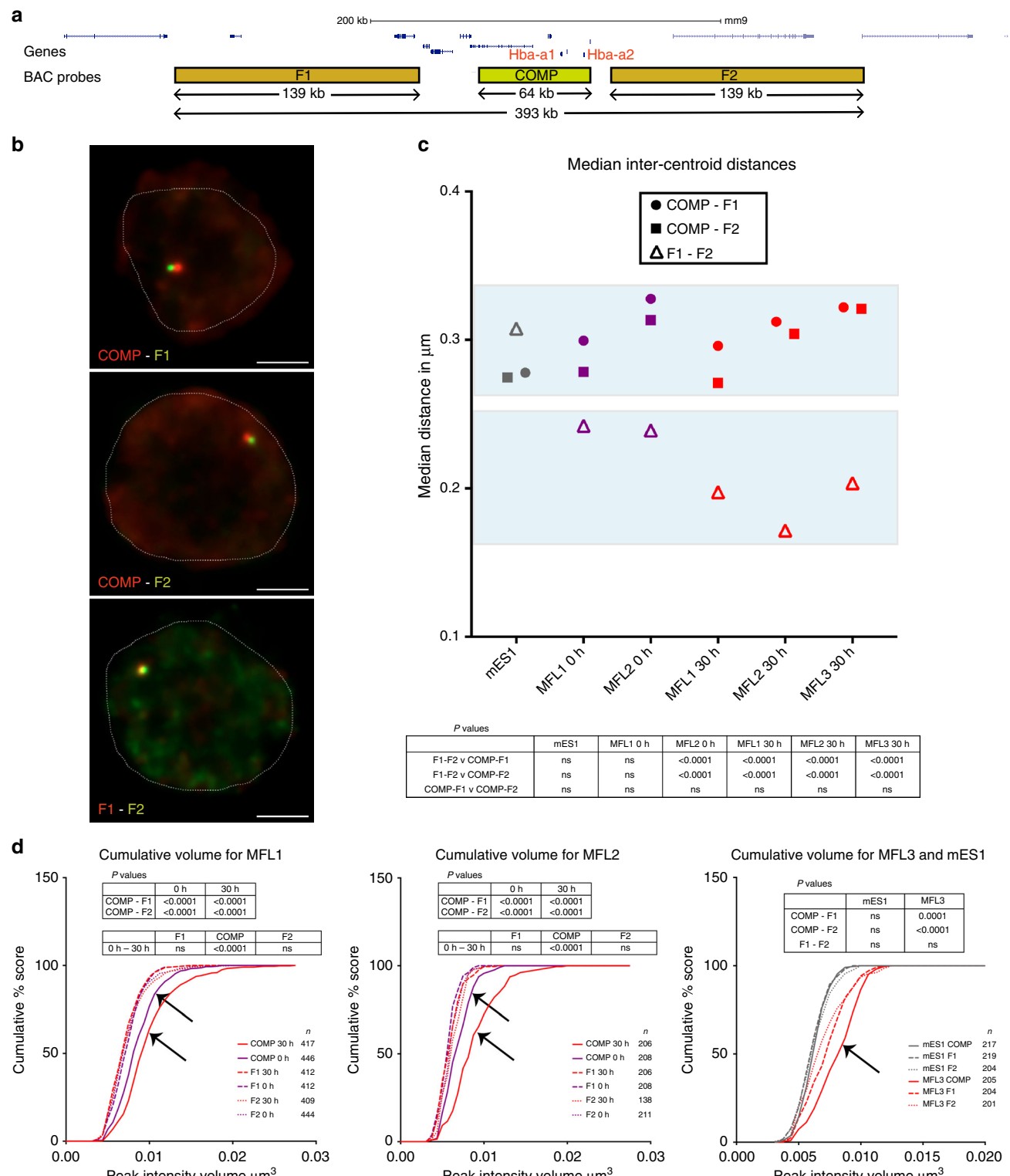

**Fig. 3** Volume and proximity measurements across the α-globin locus in WT mice. **a** Gene map and locations of the BAC FISH probes to scale, with genomic distances encompassed. **b** Representative images for the three BAC probe pairs in MFL 30 h erythroblasts with nuclei delineated (white dotted line). Scale bar 2 μm. **c** Median inter-centroid distances between the three probe pairs indicated, in mES cells (grey) and erythroblasts at 0 h (purple) and 30 h (red) timepoints. MFL1, 2 and 3 represent cultures from three individual foetal livers. Light blue shading emphasises proximity of the flanking regions F1–F2 in erythroblasts. n = 87–236 — see Supplementary Fig. 3 for the complete data set. **d** Cumulative frequency plots of BAC signal volumes in mES cells (grey) and erythroblasts at 0 h (purple) and 30 h (red). COMP values indicating expanded volume are arrowed. All P values are derived by a Kruskal–Wallis test with Dunn's multiple comparisons. ns not significant

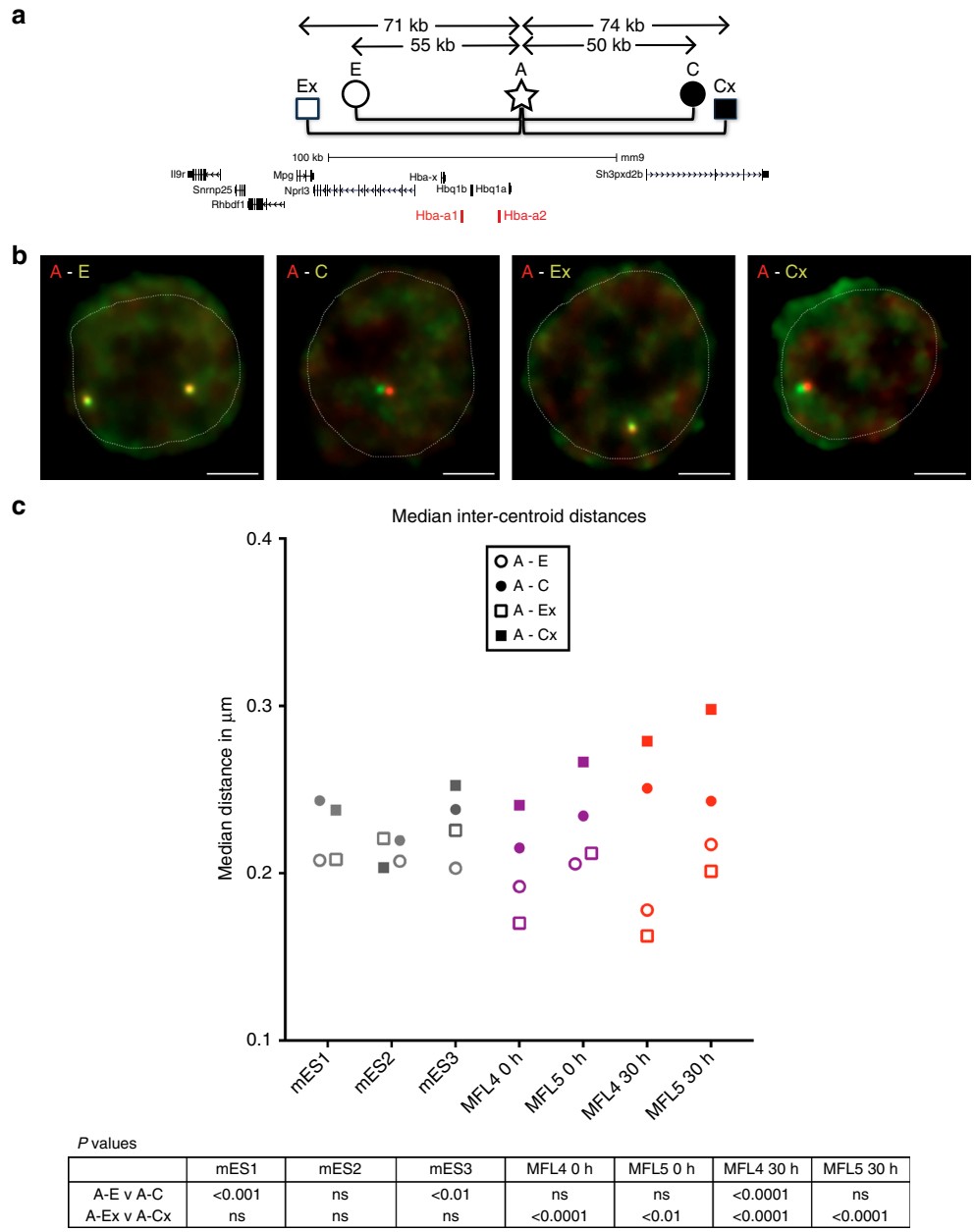

**Fig. 4** Proximity measurements at the α-globin locus in mouse WT cells. **a** Gene map with plasmid FISH probe locations showing the pairwise combinations used to measure inter-probe distances. Using probe A as a point of reference, measurements were made to the domain side (E, Ex) of probe A compared to the control non-interacting side (C, Cx). Genomic distances between midpoints of the probe pairs are shown. **b** Representative images of RASER-FISH hybridisation signals for the four plasmid probe pairs in MFL 30 h erythroblasts. White dotted line delineates nuclei. Scale bar 2 μm. **c** Median inter-centroid distances measured between the four probe pairs in three different cell types, mES1–3 (grey), MFL4–5 0 h (purple) and MFL4–5 30 h (red). p values, derived by a Kruskal–Wallis test with Dunn's multiple comparisons, are shown. See Supplementary Fig. 4 for full data with statistical analyses. At MFL 0 h and 30 h but not mES, the distance between A and Ex is consistently statistically shorter (p < 0.0001) than A to Cx

although we find no difference in the median measurements between A-E and A-C, we do find significant difference between A-Ex and A-Cx measurements, in agreement with NG Capture-C data (Fig. 2) where interaction frequencies indicate that the α-globin domain has already formed. At MFL 30 h when the α-globin genes are fully active, the difference is highly significant for A-Ex versus A-Cx, with A more frequently closer to Ex, whilst distances between A to Cx increase. These data indicate that proximity has already been established between the extremities of the self-interacting domain early in erythroid differentiation and when the majority of α-globin genes are highly active at MFL

30 h, the extremities are more frequently in close proximity, supporting the concept of a looped domain where sites A and Ex sit in regions defining the borders of generalised interactions.

**Visualising the domain with super-resolution imaging.** To visualize this region at the highest possible resolution we analysed erythroblasts using stimulated emission depletion (STED) imaging[20]. As probes, we used the COMP BAC, which precisely corresponds to the self-interacting domain and the two flanking plasmid probes (A and Ex), detecting the extremities of the domain (Fig. 5a). For 24 out of 35 loci in erythroblast nuclei, the

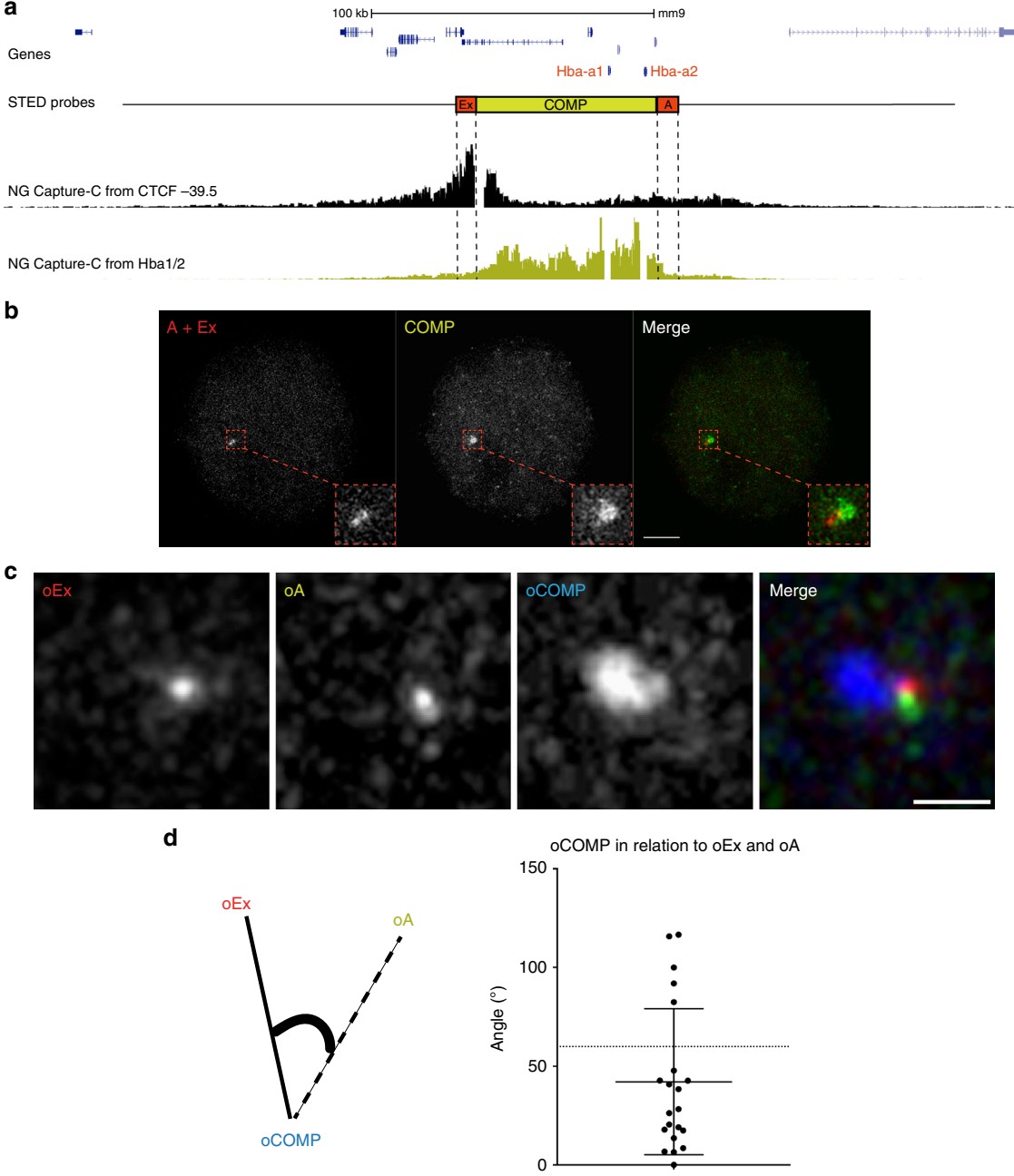

**Fig. 5** Super-resolution imaging of the α-globin domain. **a** Gene map with FISH probe Ex, COMP and A locations marked against NG Capture-C tracks depicting interactions at MFL 30 h from the viewpoints CTCF BS -39.5 (black) and *Hba1/2* promoters (lime green). **b** 2D STED maximum intensity projection images of FISH probes A and Ex (both red) which flank the α-globin domain, the COMP BAC (green) defining the extent of the domain and a merged image showing a cloud of domain signal distinct from the paired probes A and Ex. Scale bar 2 μm. **c** Three-colour 3D STED maximum intensity projection images of oligonucleotide probe pools (oEx, oCOMP and oA) designed to match the probe set shown in **a**. The oCOMP probe (blue) detects a cloud of signal that is distinct from the adjacent probes oEx (red) and oA (green). Scale bar 0.5 μm. **d** A drawing describes the angle measured in Euclidean 3D space between vectors joining the oCOMP signal with oEx and oA. The angles measured are charted on the right, with a median of 42°, well below an expected 60° for a random distribution. n = 28

signals for A and Ex were in proximity or overlapping, with the COMP signal distinctly to the side or wrapped around them (Fig. 5b). To strengthen these observations, we performed three-colour STED using pools of directly labelled oligonucleotide probes (oA, oCOMP, oEx) designed to match the three probes described above. Again, at 16 out of 21 loci in erythroblast nuclei, the two flanking probes were in proximity with a distinct signal cloud covering the active domain (Fig. 5c). By measuring the angle between vectors drawn in 3D space from centroids defining

oCOMP to oA and oCOMP to oEx, the resulting median angle of 42° further implies that the three probes are not commonly in a linear arrangement (Fig. 5d).

Based on all the data above, we present a model (Fig. 6) for the formation of an active regulatory domain at an early stage of erythropoiesis, with the self-interacting domain boundaries and external flanking chromatin frequently sited closer together and chromatin within the domain less compact relative to surrounding regions as differentiation proceeds. Notably, NG Capture-C

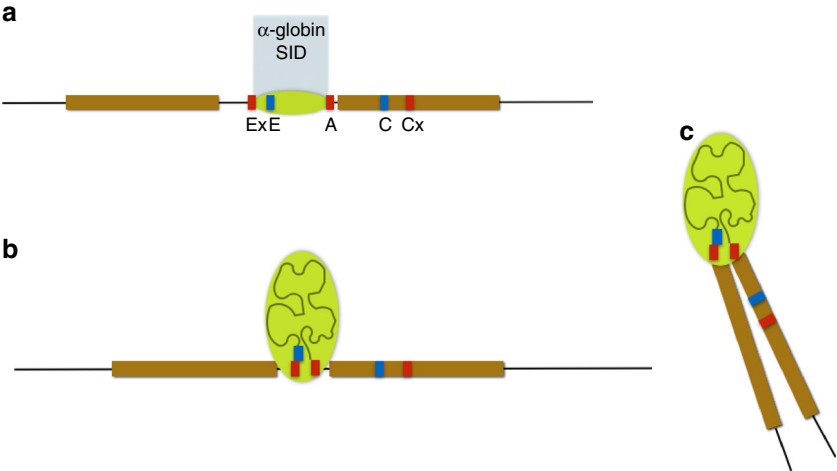

**Fig. 6** A schematic model of the alpha globin locus. Schematic model showing the α-globin self-interacting domain (SID) (lime green). Sites detected by FISH probes are as for Fig. 1. **a** represents the linear locus, while **b** and **c** depict conformations of the self-interacting domain, where the domain expands as chromatin decompacts and the flanking regions can sit in proximity

from the control sites C and Cx shows their avoidance of the α-globin domain but does detect infrequent interactions between the flanking chromatin regions (Supplementary Fig. 5), suggesting low frequency contacts of surrounding chromatin caused by formation of the domain.

**Distinguishing domain formation from internal interactions.** In engineered mice with a homozygous double knockout of the two major α-globin enhancers (DKO for MCS-R1 and MCS-R2), nascent transcription from the α-globin locus is reduced by 90%[21]. NG Capture-C analysis indicates that the self-interacting domain still forms in erythroblasts in the absence of the major enhancers when looking from boundary CTCF viewpoints (Fig. 7a). FISH inter-probe distances are in agreement with that analysis (Fig. 8a). Further, we found a highly significant overlap of F1 with F2 signals calculated by Pearson's correlation (coefficients WT 0.75 and DKO 0.67) when compared to COMP with F1 or F2 (coefficients WT 0.46 and DKO 0.46). For plasmid hybridisations, there was also a significant difference in distance measurements for both A-E versus A-C and A-Ex versus A-Cx (Fig. 8c, Supplementary Fig. 6a, b). This indicates that the structure we detect in WT erythroblasts, where the extremities of the domain come together, is recapitulated in the double enhancer knockout despite the physical absence of the core binding sites of the two enhancers and the severe reduction in transcriptional output from the α-globin genes. Notably however, by NG Capture-C interactions within the domain from the *Hba1/2* promoter viewpoints are significantly reduced (Fig. 7a). This finding distinguishes a mechanistic formation of a looped domain, independent of the internal interactions occurring within the domain related to transcription.

**Loss of α-globin genes does not affect domain formation.** We next analysed erythroblasts from a mouse homozygous for a 16 kb deletion that removes both α-globin genes from each chromosome (AMKO), consequently there is no adult α-globin transcription at all[22]. As for WT, we found a significant difference in measurements within and outside of the self-interacting domain (Fig. 8d, Supplementary Fig. 6c) and this is matched by NG Capture-C analysis, which still detects a definable self-interacting domain in the absence of the α-globin genes (Fig. 7b). Here again we see evidence that the domain structure still forms,

this time in the absence of the α-globin promoters and of transcription from the α-globin genes.

**Decompaction is independent of enhancer–promoter interaction.** We asked whether transcriptional upregulation at the α-globin locus is directly related to chromatin decompaction within the self-interacting domain. Volume measurements of the FISH signal generated using the COMP probe, together with the two flanking probes F1 and F2, indicate that the chromatin within the domain is decompacted compared to the flanking regions in both wild-type and DKO knockout cells (Fig. 8b). Hence, within the domain, chromatin decompaction is uncoupled from transcriptional upregulation of the α-globin genes, indicating a response to earlier events at the locus.

**Maintenance of domain structure.** Finally we wanted to investigate the role of ongoing transcription within the α-globin domain, both genic and non-genic, as a factor in maintaining domain structure and interactions within the domain. We knocked down global transcription in differentiating erythroblasts at MFL 30 h by a 3 h exposure to 5,6-Dichloro-1-*beta*-D-ribo-furanosylbenzimidazole (DRB), which inhibits Pol II elongation. Despite successful knockdown of transcription (Supplementary Fig. 7), the domain structure is unaltered (Fig. 9) demonstrating that overall transcriptional elongation within the domain is not contributing to maintenance of the domain structure.

**Discussion**
In contrast to many current models of long-range gene regulation, we have shown, using a synergistic combination of NG Capture-C, quantitative FISH and super-resolution microscopy, that an erythroid-specific decompacted self-interacting domain, delimited by convergent CTCF/cohesin-binding sites forms early in erythropoiesis. It has been proposed that self-interacting domains might result from specific interactions between enhancers, promoters and CTCF/cohesin elements[23–25]; however, there are no detectable tissue-specific changes in CTCF binding at the α-globin locus[17]. Recent data[17], together with the evidence presented here, show that rather than interacting directly with the α-globin enhancers and promoters, the flanking CTCF sites appear to avoid these elements: in fact, each shows interaction with the opposite flanking regions. Notably, when looking from either

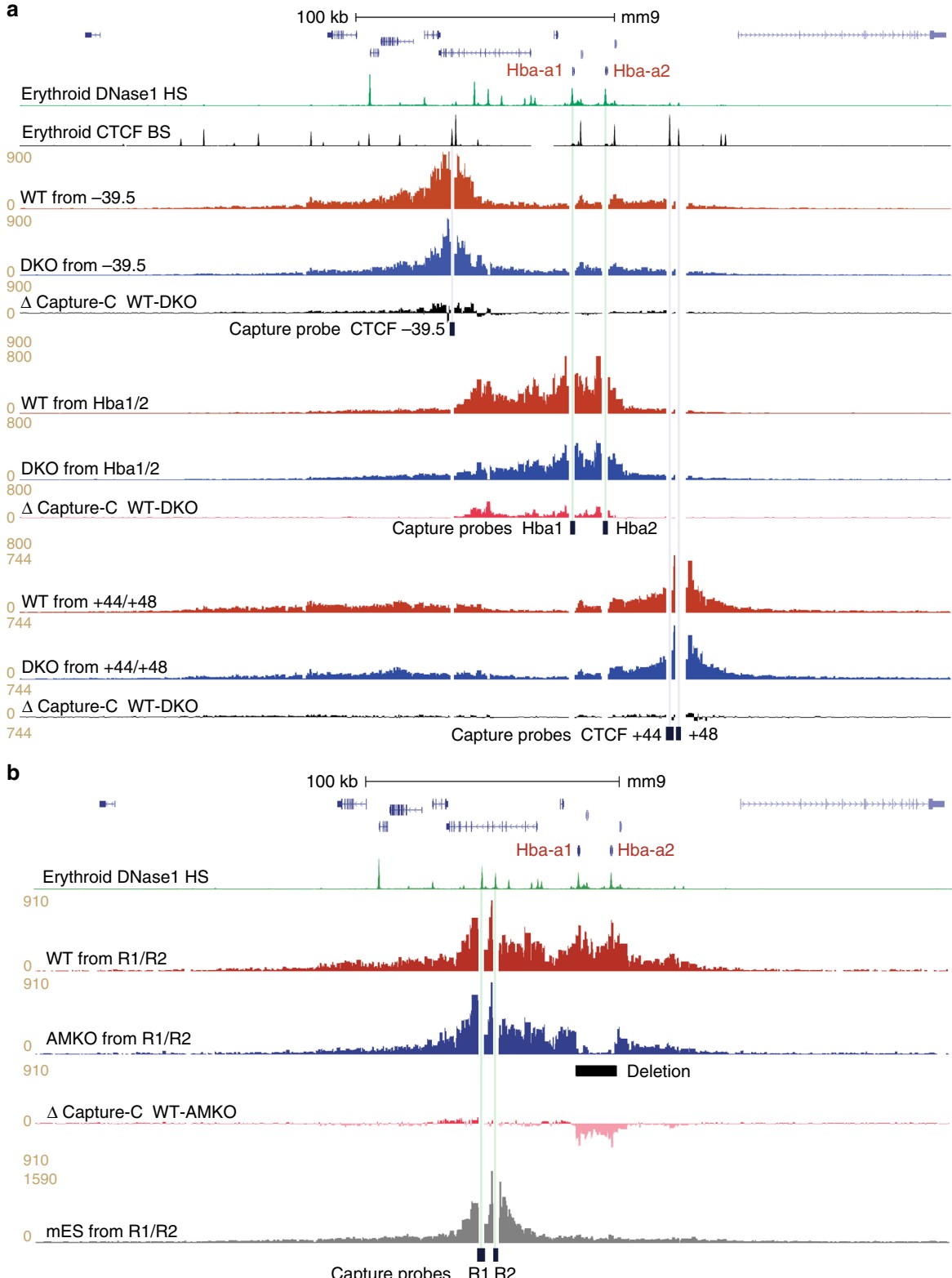

**Fig. 7** The α-globin domain still forms in the absence of elements critical for α-globin expression. **a** Gene map with DNase1 HS (green) and CTCF BS (black) genome browser tracks, followed by NG Capture-C tracks highlighting interactions from CTCF BS −39.5, *Hba1/2* and CTCF BS + 44/48 viewpoints in 30 h MFL WT (red) and MFL DKO (homozygous deletions for MCS-R1/R2) (blue), with differential tracks (Δ) showing persistence of domain structure when viewed from CTCF sites even though interaction frequencies internal to the domain are affected. **b** Gene map with DNase1 HS (green) followed by NG Capture-C tracks highlighting interactions from MCS-R1/R2 viewpoint in MFL WT (red) and MFL AMKO (blue), with a differential track (Δ) showing persistence of domain structure in AMKO when contrasted with the absence of a domain observed in mES cells (grey)

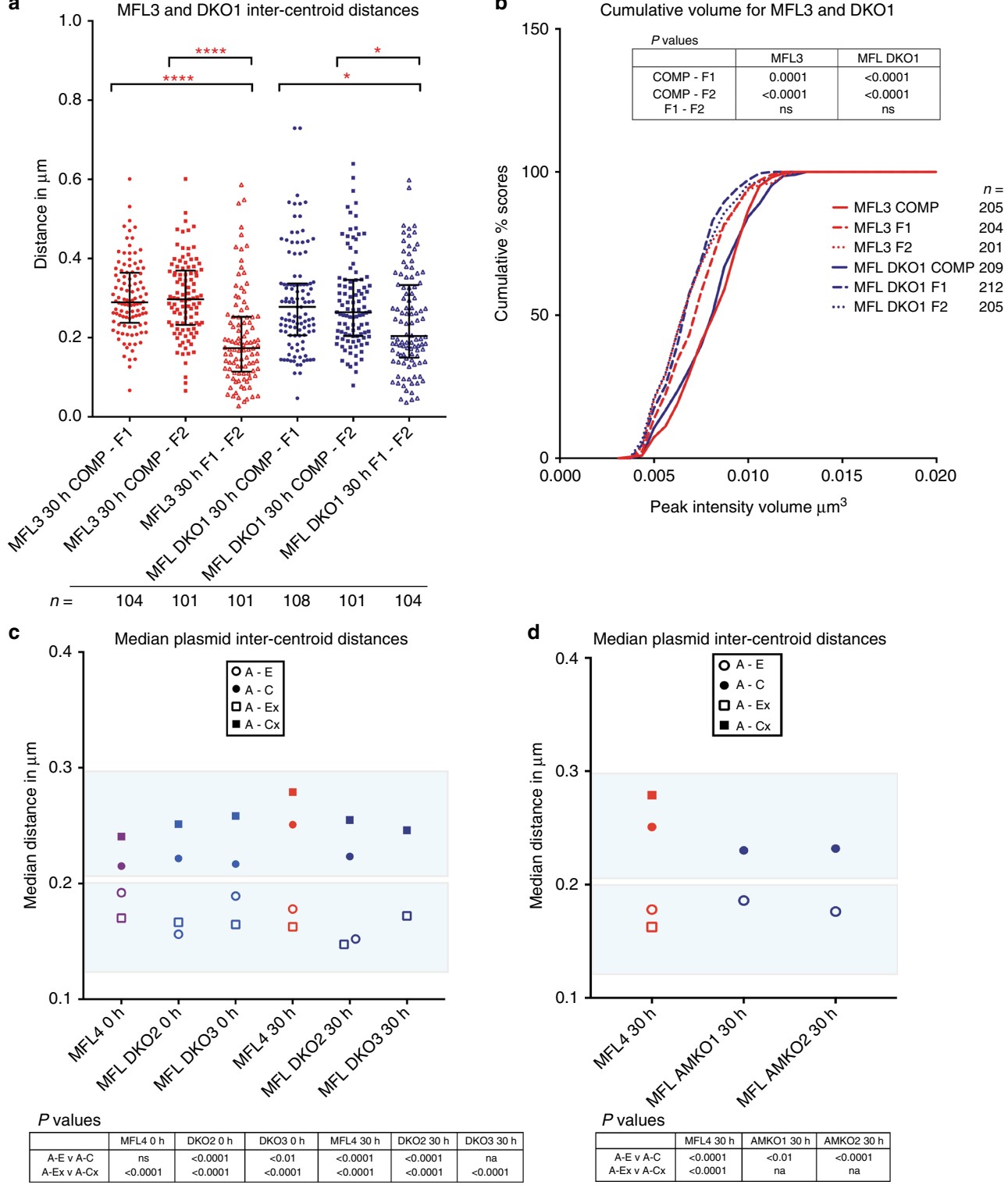

**Fig. 8** The α-globin domain still forms in the absence of elements critical for α-globin expression. **a** Pairwise inter-centroid distances between three BAC probes in 30 h erythroblasts derived from littermates MFL3 (WT) and DKO1 (homozygous deletions for MCS-R1/R2). F1–F2 are significantly closer than COMP-F1 and COMP-F2 in both WT ($p < 0.0001$ for both) and DKO1-derived erythroblasts ($p = 0.0374$ and 0.0468 respectively). Total number of measurements is indicated by 'n'. **b** Cumulative frequency plots of BAC signal volumes in 30 h erythroblasts from MFL3 and DKO1. The larger COMP volumes are arrowed. Total number of measurements is indicated by 'n'. **c** Median inter-centroid distances between four plasmid probe pairs at MFL 0 h and 30 h from littermates WT MFL4 and two homozygous double knockout embryos DKO2 and DKO3. Light blue shading emphasises the shorter distances within the self-interacting domain in both WT and knockouts. See Supplementary Fig. 6 for the complete data set. **d** Median inter-centroid distances between plasmid probe pairs A-Ex (represented as A-E distance because of a 16 kb α-globin gene deletion) and A-C at MFL 30 h in two α-globin knockout lines from littermates AMKO1 and AMKO2, plotted against WT MFL4. Light blue shading is as for **c**. See Supplementary Fig. 6 for full data. All $p$ values are derived by a Kruskal–Wallis test with Dunn's multiple comparisons. ns not significant

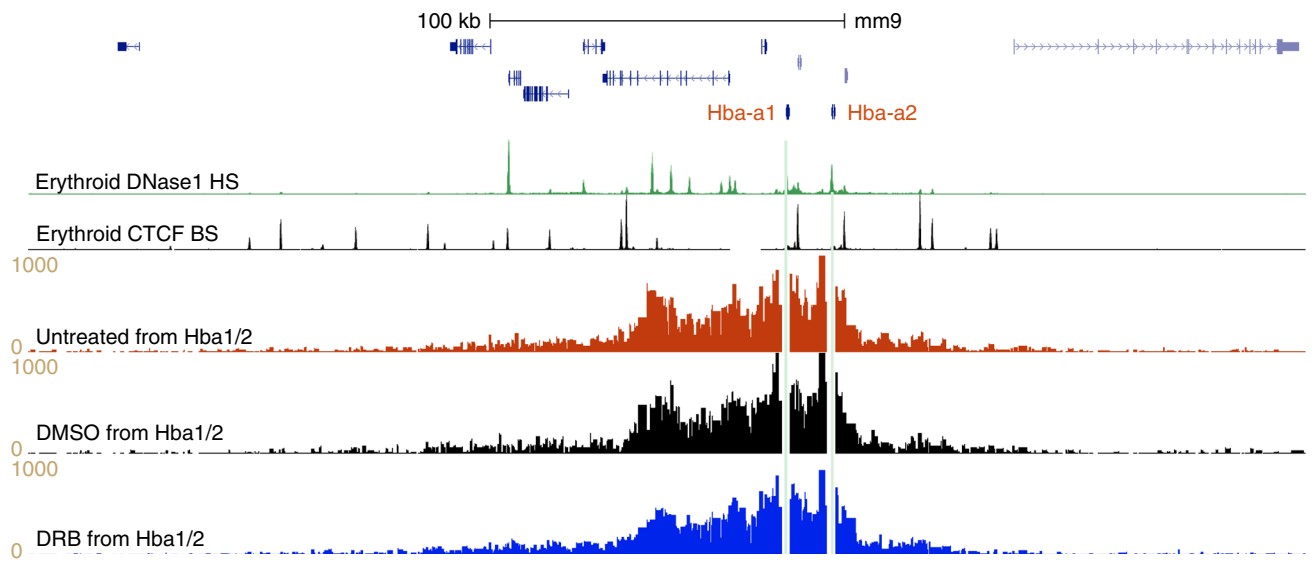

**Fig. 9** Transcriptional elongation is not required for the maintenance of domain structure. Gene map with DNase1 HS (green) and CTCF BS (black) genome browser tracks, followed by NG Capture-C tracks from the *Hba1/2* viewpoints in MFL 30 h cells untreated (red) and cultured for 3 h in presence of control DMSO (black) and DRB (blue), showing retention of interaction frequencies within the domain despite loss of transcription

boundary CTCF-binding site (−39.5 or +48), the interaction profiles do not show a punctate interaction pattern with specific CTCF sites, rather we see a broad domain of interactions with the flanking domain that is tissue specific. This suggests that the formation of the α-globin self-interacting domain has the effect of bringing the flanking regions together in a general manner (as depicted in Supplementary Fig. 5) rather than specifically tethering one CTCF-bound region to another CTCF-bound region, although this effect may be mediated by the boundary action of these sites[17]. Our findings could be explained by the recently proposed hypothesis of chromatin loop extrusion[26–28]. This is thought to be an active process in which a loop-extruding factor, containing two DNA-encompassing units such as cohesin, associates with chromatin and travels along the chromatin fibre in opposite directions, creating a progressively larger intervening loop, until the factor is stalled at appropriately orientated CTCF-bound elements. This model, although currently only a hypothesis, would explain our observations that CTCF/cohesin sites flanking the α-globin self-interacting domain become juxtaposed around a decompacted loop of chromatin in erythroblasts. The marked difference in residence times reported for cohesin and CTCF[29] implies that such loops would be repeatedly breaking and reforming, leading to a dynamic structure existing in different conformations at any one time point[30].

To investigate the role of transcription in the formation of a looped domain, we made use of two mouse mutants where levels of α-globin transcription are severely reduced (DKO) or abrogated entirely (AMKO). We show that domain formation does not rely on interactions between the α-globin genes and their enhancers, suggesting that it may more simply depend on the presence of activated lineage-specific *cis*-elements driving a transcription-independent mechanism for opening chromatin. Certainly dCas9 gene activation alone has been shown to be insufficient to create a domain structure[19], and in Drosophila more generally, TAD formation can arise independently of transcription[31]. Within a 2.7 Mb segment at the chicken α-globin locus it has also been shown that whilst genic and inter-genic transcription is upregulated in the immediate vicinity of the α-globin genes when they are active, increasing Hi-C contacts do not however automatically equate with increasing transcription[32].

By globally inhibiting Pol II activity, we further demonstrate that the self-interacting domain does not require ongoing transcriptional elongation, either genic or non-genic, for maintenance of the structure.

Dynamic chromatin decompaction at specific loci is a feature of tissue-specific and temporally controlled gene expression[33,34] although how and when this happens within the whole process of gene regulation has been unclear. By studying FISH signal volumes we detect decompaction relative to mES cells at an early stage of terminal erythroid differentiation. In the double enhancer knockout mouse (DKO) where α-globin transcription is reduced by 90%, we show that decompaction still occurs to the same degree and could therefore be uncoupled from transcriptional activity in our system. There are similar distinctions between compaction and transcription reported in different systems[35,36] although there is evidence from the β-globin locus that when active, chromatin changes shape and becomes substantially smaller in volume[37]. In live cells, both decreased[38] and increased[39] mobility of loci have been reported upon their transcriptional activation. At the α-globin locus, chromatin in an erythroid cell line (K562) was interpreted to be less compact than lymphoblastoid cells[40]. Studies in chicken[32] report compaction of the sub-TAD encompassing α-globin when this gene is fully active, however this was based on frequency of chromatin interactions alone without parallel FISH data.

In our analysis of DKO erythroblasts, we distinguished the formation of a self-interacting domain as detected by FISH and by NG Capture-C, which looked normal, from interactions within the domain, which were significantly affected. This would suggest a process for domain formation that is simply dependent on the activation of lineage-specific elements. We have previously noted an accumulation of cohesin in erythroid cells around all five enhancer-like elements of the α-globin cluster[17], which could act as entry points for cohesin, if we consider the dynamic loop extrusion hypothesis scenario. In this study, the remaining erythroid-specific elements in the absence of MCS-R1, MCS-R2 and α-globin genes are MCS-R3, MCS-R4 and MCS-Rm, which could play a redundant role in the formation of the self-interacting domain. This could suggest a potential role for *cis*-elements like MCS-R3 and MCS-Rm, which have the signature of

enhancers but without obvious enhancer activity[21], that is distinct from, and is active prior to, upregulation of gene expression. In this scenario and compatible with our data, the boundary elements of the domain would be brought into proximity as a result of loop extrusion or similar mechanism, rather than initiating the formation of a self-interacting domain.

## Methods

**Animal procedure**. The mutant and wild-type mouse strains reported in this study were maintained on a mixed background and were generated and phenotyped in accordance with Animal [Scientific Procedures] Act 1986, with procedures reviewed by the clinical medicine Animal Welfare and Ethical Review Body (AWERB), and conducted under project licence PPL 30/3339. All animals were singly housed, provided with food and water ad libitum and maintained on a 12 h light: 12 h dark cycle (150–200 lux cool white LED light, measured at the cage floor). Mice were given neutral identifiers and analysed by research technicians unaware of mouse genotype during outcome assessment.

**Cell culture**. Erythropoiesis can be faithfully recapitulated ex vivo where progenitor cells differentiate down an erythroid pathway, making all necessary proteins for red cell function, to a late stage when the nucleus condenses and is finally extruded from the cell. Ex vivo culture of foetal liver cells from E12.5 mice (MFL) allows us to access erythroblasts at an early stage of differentiation with low levels of globin transcription (MFL 0 h) and at a more differentiated stage when erythroblasts are at peak transcription of the globin genes (MFL 30 h). Differentiation status was monitored by cytospin (Fig. 1b), and level of α-globin nascent transcript was assessed by RT-qPCR (Fig. 1c) and by RNA-FISH (Fig.1d). Previous analysis of chromatin conformation at the α-globin locus has used Ter119-positive erythroblasts derived from adult spleen to represent the α-globin-on population[16,17]. Data derived from these cells and from MFL 30 h erythroblast cultures are comparable (Supplementary Fig. 1). Our in vitro mouse foetal liver (MFL) culturing system is based on previous protocols[41,42]. Briefly, MFL cells, taken at E12.5, were cultured in StemPro medium (Invitrogen) supplemented with Epo (1 U/mL) (Janssen, PL 00242/029), SCF (50 ng/mL) (Peprotech, 250-03), dexamethasone (1 μM) (Hameln, DEXA3.3) and 1× L-Glutamine (Invitrogen) for 6–7d to expand the erythroid progenitor population. Cells were differentiated, over a 30 h period in StemPro medium supplemented with Epo (5 U/mL) (Janssen, PL 00242/029) and transferrin (0.5 mg/mL) (Sigma, T0665) to a late stage of erythropoiesis. Foetal liver material was obtained from mice that are wild-type, DKO (where both MCS-R1 and MCS-R2 are deleted)[21], or AMKO (where both α-globin genes are removed)[22]. Mouse ES cell line, E14 (gift from Dr. A.G. Smith), was cultured in GMEM (Invitrogen) supplemented 10% (vol/vol) FBS (Gibco®, 10270) and LIF and grown in gelatinised flasks. C127, a mouse mammary epithelial cell line (gift from Dr L. Schermelleh), was cultured in DMEM (Invitrogen) supplemented with 10% (vol/vol) FBS (Sigma), 1× penicillin/streptomycin (Invitrogen) and 1× L-glutamine (Invitrogen). MEL, the mouse erythroleukaemia cell line (gift from Dr A. Deisseroth), was cultured in RPMI (Invitrogen) supplemented with 10% (vol/vol) FBS (Sigma), 1× penicillin/streptomycin (Invitrogen) and 1× L-glutamine (Invitrogen). Mouse embryonic fibroblasts (MEF) (generated in-house) were cultured in DMEM (Invitrogen) supplemented with 15% (vol/vol) FBS (Gibco®), 1× penicillin/streptomycin (Invitrogen), 1× L-glutamine (Invitrogen) and 1× NEAA (Invitrogen). All cells were incubated at 37 °C in a humidified 5% (vol/vol) CO$_2$ incubator. None of the cell lines used here are found in the database of commonly misidentified cell lines that is maintained by ICLAC and NCBI Biosample and all are routinely screened for mycoplasma infection.

**Fluorescence activated cell sorting and flow cytometry**. Defined cell populations were sorted as follows; expanded MFL cells were depleted of differentiated erythroid Ter119+ve cells by staining with Ter119 antibody 1/100 (Becton Dickinson, 553673) and separation using MACS column (Miltenyi Biotec Ltd). Ter119 −ve cells were then stained and sorted for CD44 1/100 (Becton Dickinson, 561862) and cell size (Fig. 1a). This gave an early erythroid progenitor population. Following 30 h culturing in differentiation medium, more differentiated erythroblasts were obtained. Progression of in vitro differentiation was monitored by flow cytometry, by staining with Ter119 antibody 1/100 (Becton Dickinson, 553673).

**Reverse transcription qPCR**. For MFL characterisation, isolation of total RNA was performed by lysing 10$^7$ cells in TRI reagent (Sigma), according to the manufacturer's instructions. To remove genomic DNA from RNA samples, samples were treated with TURBO™ DNase according to manufacturer's protocol (Invitrogen, AM2238). To assess relative changes in gene expression by qPCR, 1 μg of total RNA was used for cDNA synthesis using Superscript™ II reverse transcriptase (Invitrogen, 18064014). Quantification of mRNA levels was performed in triplicate using SYBR® Green Real Time PCR master mix according to manufacturers instructions (Applied Biosystems, 4309155). The relative standard curve method was used for relative quantitation of RNA abundance. For transcriptional inhibition, RNA samples were prepared using Direct-zol RNA Miniprep kit (Zymo Research, R2050). 50 ng of total RNA was used for cDNA synthesis using

Superscript III™ First Strand Synthesis Super-Mix (Invitrogen, 11752-050) and quantification was performed in triplicate using Fast SYBR™ Green Master Mix (Applied Biosystems, 4385612). The 2$^{-\Delta\Delta Ct}$ method was used to calculate the relative change in transcription. Primers: mHba Nascent F (GTGTGGATCCCGTC AACTTC); mHba Nascent R (CCACTATGTTCCCTGCCTTG); mHbb Nascent F (GCACCTGACTGATGCTGAGA); mHbb Nascent R (TCTCCAAGCACCCAAC TTCT); m18s F (GTAACCCGTTGAACCCCATT); m18s R (CCATCCAATCGG TAGTAGCG).

**Next-generation Capture-C**. Performed as previously described[16]. Material was obtained from mES E14 cells and MFL cells (0 h and 30 h) from WT and AMKO. Briefly, 3 C libraries were generated using standard methods. Before oligonucleotide capture, 3 C libraries were sonicated to a fragment size of 200 bp and Illumina paired-end sequencing adaptors (New England BioLabs, E6040, E7335 and E7500) were added using Herculase II polymerase (Agilent). Samples were indexed, allowing multiple samples to be pooled before oligonucleotide capture using biotinylated DNA oligonucleotides designed for the α-globin gene promoters Hba1/2, the MCS-R1 and −R2 regulatory elements, CTCF −39.5 (Sigma Aldrich) and the five FISH probe sites Ex, E, A, C, Cx (ATDBio Ltd). The first hybridisation reaction was scaled up relative to the number of samples included in the reaction to maintain library complexity using Nimblegen SeqCap EZ Hybridization and Wash Kit (Roche, 05634261001). After a 72 h hybridisation step, streptavidin bead pulldown (Invitrogen, 65305) was performed, followed by multiple bead washes using Nimblegen SeqCap EZ Hybridization and Wash Kit (Roche, 05634261001) followed by PCR amplification of the captured material using SeqCap EZ accessory Kit v2 (Roche, 07145594001). A second capture step was performed as above, with the exception that it was carried out in a single-volume reaction. As material was limited in the NG Capture-C experiments after treatment with DRB, these were performed with modifications optimized for small cell numbers[43], according to the Low-Input Capture-C protocol[44]. The material was sequenced using the Illumina® MiSeq platform with 150-bp paired-end reads. Data were analysed using scripts available at https://github.com/Hughes-Genome-Group/CCseqBasicF/releases and R was used to normalize data and generate differential tracks.

**Probes and nick translation labelling**. For FISH; plasmid probes used were pEx (mm9; chr11; 32129812-32136918), pE (mm9; chr11; 32146280-32153457), pA (mm9; chr11; 32201016-32208529), pC (mm9; chr11; 32251235-32258747) and pCX (mm9; chr11; 32275986-32282385). Probes were constructed in the pBlue-Script plasmid by subcloning regions from mouse BAC RP23-469I8 and BAC RP24-278E18 by λ-red-mediated recombination[45]. Mouse BACs were as follows; F1 (RDB 4214 MSMg01-530C17), COMP (RDB 4214 MSMg01-276J20) engineered by λ-red-mediated recombination to give a final insert size covering mm9; chr11; 32137046-32200781, and F2 (RP24-278E18). All BACs were obtained from RIKEN[46] and BACPAC Resources Center (Children's Hospital Oakland Research Institute; [https://bacpacresources.org/]). FISH probes were labelled by nick translation as previously described[47]. Probes were directly or indirectly labelled by nick translation using Cy3-dUTP (GE Healthcare) and digoxigenin-11-dUTP (Roche). Oligonucleotide probes were designed as 50mers, tiled across the region of interest on both strands. They were generated by microarray synthesis and amplified by circle-to-circle (RCA) amplification according to the method described by Schmidt et al[48]. Amino-11-dUTP (3:1 with TTP) was incorporated during RCA and then labelled with fluorophore NHS esters (Abberior Star Red, Atto565 and Oregon Green 488). Conditions for labelling: 1 mM oligonucleotides, 10 mM NHS ester (added as 0.1 M solution in DMSO), 0.5 M sodium carbonate buffer tabfigpH 8.5, shaken at 55 °C for 5 h and purified by gel filtration.

**RASER-FISH**. The small size of the locus requires optimal preservation of 3D nuclear structure. However, conventional FISH requires heat denaturation disrupting fine details of chromatin structure below 1 Mb[49,50]. Here we have successfully adapted the principle of chromosome orientation FISH (CO-FISH)[51], to non-repetitive genomic loci. The resulting RASER (resolution after single-strand exonuclease resection)-FISH method maintains nuclear fine-scale structure by replacing heat denaturation with exonuclease digestion, and is suitable for high- and super-resolution imaging analysis. Line profiles across DAPI-stained nuclei after three separate treatments (immunofluorescence (IF), our standard 3D-FISH and RASER-FISH) indicated a loss of structure in the 3D-FISH nuclei that is not observed in the RASER-FISH nuclei when compared to IF only (Supplementary Fig. 8). When comparing RASER-FISH to 3D-FISH, hybridisation efficiency was similar for the two techniques (>90%), suggesting that exonuclease digestion around the alpha globin locus is extensive by this method. Briefly, cells were labelled overnight with BrdU/BrdC mix (3:1) at final conc. of 10 μM. Cells were fixed in 4% PFA (vol/vol) for 15 min and permeabilised in 0.2% Triton X-100 (vol/vol) for 10 min. Cells were then stained with Hoechst 33258 (0.5 μg/mL in PBS), exposed to 254 nm wavelength UV light for 15 min, then treated with Exonuclease III (NEB) at final conc. 5 U/μL at 37 °C for 15 min. Labelled probes (100 ng each) were denatured in hybridization mix at 90 °C for 5 min, BACs were preannealed at 37 °C for 20 min. Coverslips were hybridized with prepared probes at 37 °C either overnight or for 50 h for the oligonucleotide probes. After hybridization, coverslips were washed for 30 min twice@@ in 2× SSC at 37 °C, once in 1 × SSC at RT.

Coverslips were blocked in 3% BSA (wt/vol) and digoxigenin was detected with sheep anti-digoxigenin FITC 1/50 (Roche, 11207741910) followed by rabbit anti–sheep FITC 1/100 (Vector Laboratories, FI-6000). Coverslips were stained with DAPI (0.5 μg/mL in PBS), washed with PBS and mounted in Slowfade® Diamond mountant for standard widefield imaging (Molecular Probes®) and either Slowfade® Diamond or Vectashield (Vector Laboratories) for STED imaging.

**Standard 3D DNA-FISH.** Cells were fixed in 4% PFA (vol/vol) for 15 min and permeabilized in 0.2% Triton X-100 (vol/vol) for 10 min. Cells were denatured in 3.5 N HCl for 20 min and neutralized in ice-cold PBS. Probes were prepared as in the previous section, and coverslips were hybridized overnight at 37 °C. Cells were washed and blocked, probes were detected and coverslips were mounted as in the previous section.

**Tolerance.** Pools of oligonucleotide probes were designed consisting of 30 nt tiling 6 kb of the MCS-R2 region, avoiding large repeats, with 30 nt gaps between probes (80 oligonucleotides in total). The probes were synthesised with 5'-amino groups using standard phosphoramidite chemistry (ATDBio Ltd). After purification by gel filtration, the probes were labelled in pools covering 1 kb with either digoxigenin NHS ester or Cy3 NHS ester, to give a 6 kb probe with alternating 1 kb regions of Cy3 or digoxigenin. Conditions for labelling: 1 mM oligonucleotides, 10 mM NHS ester (added as 0.1 M solution in DMSO), 0.5 M sodium carbonate buffer pH 8.5, shaken at 55 °C for 5 h and purified by gel filtration followed by RP-HPLC eluting with a 0.1 M TEAA/MeCN gradient. Fractions containing the products were combined, dried, desalted by gel filtration and lyophilised. MEL cells were fixed on coverslips and prepared according to the RASER-FISH protocol. The pooled probes were resuspended in water at 100 ng/ μL. 1 μL of the labelled oligonucleotide mixture was added to 5 μL hybridisation buffer (Kreatech) and 5 μL 2× SSC. The probe mixture was denatured at 95 °C for 5 min, placed on ice, then applied to the coverslip. The coverslips were hybridized at 37 °C overnight, then washed, detected and mounted as previously described.

**RNA-FISH.** In brief[52], cells were fixed in 4% PFA (vol/vol), 5% (vol/vol) glacial acetic acid in normal saline for 20 min and treated with 0.02% pepsin (wt/vol) in 0.01 N HCl for 5 min. After a brief post fixation in 4% PFA, slides were dehydrated through an ethanol series. Hapten-labelled oligonucleotide probes (30 ng per slide) were resuspended in hybridization buffer and slides were hybridized overnight at 37 °C. Slides were washed in 2× SSC and following blocking, probes were detected and coverslips were mounted as described in subsection 'RASER-FISH'.

**Imaging equipment and settings.** Widefield fluorescence imaging was performed at 20 °C on a DeltaVision Elite system (Applied Precision) equipped with a 100 ×/ 1.40 NA UPLSAPO oil immersion objective (Olympus), a CoolSnap HQ2 CCD camera (Photometrics), DAPI (excitation 390/18; emission 435/40), FITC (excitation 475/28; emission 525/45) and TRITC (excitation 542/27; emission 593/45) filters. 12-bit image stacks were acquired with a z-step of 150 nm giving a voxel size of 64.5 nm x 64.5 nm x 150 nm. Image restoration was carried out using Huygens deconvolution Classic Maximum Likelihood Estimation (Scientific Volume Imaging B.V.). STED images were acquired at 20 °C on a Leica TCS SP8 3X Gated STED (Leica Microsystems), equipped with a pulsed supercontinuum white light excitation laser at 80Mhz (NKT), and two continuous wavelength STED lasers at 592 nm and 660 nm. HyD detectors were used in gated mode (1.5–6 ns for 592 depletion and 0.5–8.5 ns for 660 depletion). A sequential imaging mode was set employing first the 660 nm STED laser, and then the 592 nm STED laser to give a final voxel size of 31.9 nm x 31.9 nm x 110 nm in the image shown (Fig. 5b), which was minimally smoothed by performing a Gaussian blur of 0.75 pixel radius in ImageJ [https://imagej.net/]. For three colours, 3D gSTED images were collected using two continuous wavelength STED lasers at 592 nm and 660 nm and a pulsed laser at 775 nm to give a voxel size of 41.2 nm x 41.2 nm x 108 nm (Fig. 5c). Images for display were minimally smoothed in LAS X using a kernel size of 5.

**Image analysis.** Measurements of either distance or volume were made using in-house scripts [https://github.com/dwaithe/foci_measurements] [DOI: 10.5281/zenodo.1318160] in ImageJ. As a pre-processing step image regions are chromatically corrected to align the green and the red channel images. The parameters for the chromatic correction were calculated through taking measurements from images of 0.1 μm TetraSpeck® (Molecular Probes®) and calculating the apparent offset between images in each colour channel. Cells were only selected for analysis where there was no hint of replicated signal. Since the a-globin locus replicates very early in S phase, the majority of cells analysed were in the G1 phase of the cell cycle. For both distance and volume measurement scripts, signal pairs were manually identified whereupon a 20 × 20 pixel and 7 x z-step sub-volume was generated centred on the identified location (Supplementary Fig. 9a). In each identified region, thresholding was applied to segment the foci. Firstly the image region was saturated beyond the top 96.5 % intensity level, to reduce the effect of noisy pixels, and then the threshold was calculated as being 90 % of the maximum intensity value of the processed image. This was repeated for both green and red channels and was found to accurately segment the foci from background. Once segmented, signal centroid positions were mathematically calculated and the inter-centroid 3D

distance measurement was output along with a.png image for visual inspection (Supplementary Fig. 9a). For the volume analysis, the segmented volume for foci was integrated and converted into μm³ units and output for each signal. We validated any increase in volume between MFL 0 h and 30 h by taking volume measurements of fluorescently labelled 500 nm diameter Tetraspeck® beads (Molecular Probes®) incorporated into the mountant where we found the bead volume measurements equivalent at the two timepoints. Correlation of the positioning of paired FISH probes was assessed by Pearson co-efficient of correlation analysis and was performed on the 20 × 20 × 7 raw intensity signals from each channel. Line profile analysis was performed using the Plot Profile function in Fiji[53]. We made initial comparisons between z-steps of 100 nm, 150 nm and 200 nm to assess any effect on inter-centroid 3D distance measurements (Supplementary Fig. 9b) and established the tolerance of the inter-centroid distances produced by the analysis pipeline to be 53 nm (Supplementary Fig. 9c). The oligonucleotide probe STED images were analysed in Imaris (Bitplane AG, Zurich, Switzerland); elongated spots were fitted and the angles between them trigonometrically calculated from their coordinates using the law of cosines ($c^2 = a^2 + b^2 - 2ab\cos\theta$).

**Transcriptional inhibition.** MFL 30 h cells were either untreated, or treated with 100 μM 5,6-dichlorobenzimidazole 1-β-D-ribofuranoside (DRB) dissolved in DMSO or DMSO alone (control) for 1 h and 3 h in standard tissue culture conditions. RNA samples for RT-qPCR and chromatin for NG Capture-C were taken for subsequent analysis.

**Statistics and reproducibility.** Samples sizes were not predetermined. For FISH the sample sizes compare very favourably with other reported FISH studies. Full FISH data in its entirety for all biological replicates are shown in Fig. 3, 8, and Supplementary Figs. 3, 4, 8. The number of individual measurements made in each sample is given as 'n'. Statistical analysis was carried out with Graphpad Prism (version 7.0c) unless otherwise indicated. Gene expression experiments (Fig. 1c) were performed on three biological replicates (standard deviation (s.d.) is shown). All NG Capture-C experiments were performed on three biological replicates with the exception of WT and AMKO capture from MCS-R1/R2, and transcription inhibition, which were each derived from one sample. The standard deviation was calculated in R and statistical significance in the differential analysis was calculated using DESeq2 as previously described[16]. All graphs showing FISH signal inter-distance data display median values with variance shown as the interquartile range with the exception of Supplementary Fig. 9c, which show mean values with s.d. All volume analyses are displayed as cumulative frequency plots where the bins were in voxel sized increments. The statistical significance of differences in the range of distance measurements and volume measurements were derived as two-tailed by the non-parametric Kruskal–Wallis test with Dunn's multiple comparisons. $p$ values are represented as *$p < 0.05$; **$p < 0.01$; ***$p < 0.001$; ****$p < 0.0001$.

**Code availability.** The code used to analyse NG Capture-C data can be found at (https://github.com/Hughes-Genome-Group/CCseqBasicF/releases). Analysis scripts for distance and volume measurements are available at (https://github.com/dwaithe/foci_measurements) (https://doi.org/10.5281/zenodo.1318160).

## Data availability

NG Capture-C data that support the finding of this study have been deposited in the Gene Expression Omnibus (GEO) under accession code GSE107675. Statistics analyses codes are available upon reasonable request. All image files are archived in OMERO[54] and can be made available upon reasonable request. Any other data supporting the findings of this study are available from the corresponding author on reasonable request.

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

## Acknowledgements

We thank S. Butler for tissue culture support, J. Sloane-Stanley and J. Sharpe of the Transgenics Facility for mouse breeding and foetal liver provision, K. Clark and C. Waugh of the Flow Cytometry Facility for cell sorting services, E. Repapi for advice on statistical analysis, E. Garcia of the Wolfson Imaging Centre for advice on STED imaging, J. Davies and C. Harrold for provision of libraries and analysis of NG Capture-C data, T. Brown for oligonucleotide synthesis support and ATDBio Ltd. for generous provision of oligonucleotides. We wish to acknowledge the Computational Biology Research Group, MRC Weatherall Institute of Molecular Medicine, Radcliffe Department of Medicine and University of Oxford for use of their services in this project. The facility is supported by the MRC Strategic Award to the institute. This work was supported by the Medical Research Council (MC_UU_12009 to V.J.B., D.H., J.H. and MR/N00969X/1 to J.H.) and Wellcome Trust (106130/Z/14/Z to DH). Further support came from grants to the Wolfson Imaging Centre Oxford (Wolfson Foundation 18272, joint MRC/BBSRC/EPSRC MR/K015777X/1, Wellcome Trust Multi-User Equipment 104924/Z/14/Z) and the WIMM FACS Core Facility (NIHR Oxford BRC and John Fell Fund (131/030 and 101/517), the EPA fund (CF182 and CF170) and by the WIMM Strategic Alliance awards G0902418 and MC_UU_12025.

## Author contributions

V.J.B. and D.R.H. conceived the project. J.M.B. and V.J.B. developed the RASER-FISH technique from Co-FISH and performed the FISH experiments with assistance from I.S. C.B. sub-cloned the FISH probes and S.D.O. synthesised oligonucleotides. J.M.B. and S. D.O. performed the STED imaging and associated analysis. C.L. assisted with the imaging and image storage. D.W. wrote scripts for and advised on image analysis. A.M.O. designed analysis of the α-globin domain by NG Capture-C and performed the transcription inhibition capture. N.R. and J.T. undertook the NG Capture-C experiments and subsequent analysis respectively. B.G. and M.T.K. developed the erythroblast ex vivo differentiation system and B.G. performed the nascent transcript quantification. FACS

and flow cytometry was undertaken by B.G. and C.S. V.J.B., J.M.B., J.R.H. and D.R.H. wrote the paper.

## Additional information

**Competing interests:** The authors declare no competing interests.

