## [Peer Review File · Nature Communications]

Reviewers' comments:

Reviewer #1 (Remarks to the Author):

Brown et al. used both imaging and 3C omics approaches to study local organization at the α -globin locus during early erythropoiesis in mouse cells cultured in vitro. Studies that wed these experimental approaches can be quite powerful and with careful data analysis can provide more than simple cross-validation. Especially if live cell or super resolution microscopy is used, novel and unexpected insights are sometimes obtained. Brown et al. performed NG-Capture C analysis (3C with anchors flanking or within the locus) to obtain local proximity data at 0 and 30 hrs of erythroid differentiation in vitro and correlated this with changes in median distance FISH measurements between various regions within and immediately flanking the locus. The authors used a variation of co-FISH to avoid denaturation during DNA FISH and distances were measured from 3D images obtained using standard diffraction limited microscopy. In one case, they also used STED (stimulated emission depletion) imaging to map the 3D disposition of the open and closed locus with super resolution accuracy.

General comments:

1. The title does not describe a novel result. In the last few years, it has become clear that facultative regions are generally open before transcription is maximal (e.g. Therizols et al., 2014, *Science* 346:1238; Hug et al., 2017, *Cell* 169:216). The title should state that this paper focuses on correlating imaging and omics data to study the local structure of the mouse α -globin locus. Results should be compared and contrasted to Ulianov et al., 2017 *Epigenetics & Chromatin* 10:35, which describes local chromatin changes at the α -globin locus during erythrocyte development in chicken.
2. The large spread of distance measurements does not allow the authors to conclude that the region is dynamically changing shape (lines 153-154). Differences between cells in the population, such as position in cell cycle, or degree of differentiation, would give similar variation in distance measurements. There are also technical considerations, see #4 under this heading and #3 under specific comments below.
3. Several publications have shown that proximity data correlates best with very short distance data, rather than the median distance. See Giorgetti and Heard, 2016, *Genome Biol* 17:215; Dekker, 2016, *Nat Rev Mol Cell Biol* 17:741; Fudenberg and Imakaev M, 2017, *Nat Methods* 14:673. It is unclear whether some discrepancies in the data would be resolved if the data analysis approach took this into account. See also #4.
4. The small, often sub-diffraction changes in distance between elements in local gene structure are best measured using super resolution microscopy. This paper would be quite powerful if FISH results were analyzed using STED. The STED panel presented in Figure 3 is provocative, but results using oligo FISH probes sampling the whole locus should be presented. This would go a long way toward substantiating the model and data in Extended Figure 5.
5. The loop extrusion model should be presented as speculative.
6. There is no evidence presented that any cis elements (including any enhancers or promoters) are involved with local structure changes. This conclusion should be removed from the discussion.

Specific comments:

1. The transcriptional analysis could be clarified and expanded. A) Another repeat of the α -globin RT-qPCR would help - the error bars in Figure 1b are so large that it appears there is no difference in transcription between 6 and 24 hours. B) What happens with transcription outside of the α -globin locus?

2. TER119 levels should be shown at 0 and 30 hrs and cytopsin results presented to help authenticate the in situ differentiation. The manuscript text should be reviewed and references to 30 hr cells as intermediate erythroblasts should be deleted.

3. In the Image Analysis section under Methods, it is stated that signal centers were chosen by hand. It is more accurate to identify the center of maximum intensity mathematically, rather than by eye. Additionally, many BAC and oligo probes do not give spherical signal, rather it can be elongated or lobated. Centroid identification needs to take this into account.

4. Only one CTCF site was used as an anchor for the 3C experiments. The downstream flanking site should also be used. Why is there no peak at this second site showing interaction with the CTCF-39.5 in Figure 3a?

Reviewer #2 (Remarks to the Author):

Brown et al. investigate a self-interacting chromatin domain containing the mouse alpha globin locus and delimited by CTCF/cohesin binding sites. Elements within this domain interact with high frequency with other loci throughout this domain, but not outside it. These interactions are detected in mouse fetal liver cells (MFL), which are highly enriched for erythroid lineage cells, but not in ES cells., as judged by NG Capture-C.

The authors develop a technique called Raser FISH, which preserves chromatin conformation. They use Raser FISH with a series of probes within and adjacent to the alpha globin self-interacting domain, assessing two parameters: inter-probe distances, and the volumes occupied by each probe. They use both large, BAC-derived probes, and shorter plasmid-derived probes, and examine these in ES cells, and in MFL harvested from E12.5 embryos ('MFL 0h') or differentiated in vitro for 30 h ('MFL 30h'). They find that the distance between F1 and F2, two probes flanking the domain, decreases in MFL, and is lowest in MFL 30h. Further, the median volume of the occupied by the COMP probe, corresponding to the entire domain, is larger than the F1 and F2 volumes and enlarges with differentiation, whereas the volume of all three probes is comparable, and low, in ES cells. These and studies with the shorter plasmid probes are consistent with a loop-extrusion model in which CTCF/Cohesin brings together the domain's boundaries, forming a de-compacted loop.

The authors then go on to examine MFLs derived from mice in which two intra-domain major enhancers, usually required for alpha globin expression, are deleted. They also examine mice deleted for the two alpha globin genes. In all these cases, they find that the differentiation-dependent reconfiguration of the alpha globin domain is unperturbed.

On the strength of these findings, the authors conclude that

1. The alpha globin locus forms a self-interacting domain in a tissue-specific and developmental-stage-specific manner
2. The configuration of chromatin at this domain is consistent with the loop extrusion model
3. alpha globin transcription is not a pre-requisite for the looping of the domain or its decompaction.

The authors speculate that non-enhancer cis-acting element(s) that bind tissue-specific transcription factors and that potentially do not contribute to gene transcription may be responsible for domain looping/ decompaction, a suggestion that presumably they will examine in future work.

The Raser FISH, high resolution microscopy analysis and the NG capture -C data are all performed with

high technical rigor in well-designed experiments, and the data is convincing and elegant. The conclusion that domain looping and decompaction is upstream of gene transcription is especially interesting.

My only reservation relates to the dynamics described for the chromatin reconfiguration events. Throughout the manuscript, the authors repeatedly suggest a gradual process, where the activated configuration of the alpha globin domain is found with increasing 'frequency' in later stages of development. This description would only hold if the authors had performed their experiments with MFLs at well delineated developmental stages. The authors state that MFL 0h do not undergo alpha globin transcription, whereas MFL 30h are at peak alpha globin transcription; neither is likely to be entirely correct. Both MFL 0h and MFL 30h are likely to be mixtures of non-transcribing and transcribing cells, albeit at different proportions. (Indeed, the single cell inter-centroid distance data has a much larger range in MFL 30h than in ES cell, consistent with MFL 30h being a mixture of two populations, see Extended Data Figure 2). For MFL 0h, the authors select CD44^{high} Ter119^{neg} cells, which are likely to be a mixture of CD71^{high} and CD71^{med/low} cells; the former would most likely already contain the active chromatin conformation (see Pop et al., PLoS Biology 2010) and are likely to be undergoing alpha globin transcription, though the transcript may not yet reach the threshold required for detection. Similarly, MFL 30h might contain some residual cells that have not yet differentiated. It is therefore not possible to make any statements regarding the dynamics of the chromatin conformational change with this specific dataset. This issue could easily be addressed, though, by re-stating this problem upfront and focusing on the principal findings in this manuscript, which are not affected by this issue.

Reviewer #3 (Remarks to the Author):

The authors investigate the occurrence of the "self-interacting domain," as defined in prior studies by NG Capture-C analysis, at the murine alpha-globin locus in cultured, differentiating primary fetal erythroblasts. They determine that the domain appears at both the 0 hr. and 30 hr. timepoints, while alpha-globin transcription is significantly upregulated over the same timeframe. They then perform FISH analysis using BAC probes corresponding to the self-interacting domain (COMP) or to flanking regions; this shows that while relative distances between COMP and the flanking region probes are similar in ES cells and 0 or 30 hr. erythroblasts, the distances between the two flanking probes appear to shorten in 0 hr. erythroblasts, and then shorten further in 30 hr. erythroblasts. Meanwhile, the volume occupied by the COMP probe is greater than that occupied by the flanking probes in erythroblasts, suggesting a less compact structure. FISH with smaller plasmid probes within the region appears similarly to show an increase in the distance between a probe within the domain and probes outside of it in erythroblasts compared to mESCs, while distances within the domain decrease. High resolution STED imaging shows that the plasmid probes corresponding to the extremities of the alpha-globin domain overlap in many cells, while the COMP BAC probe corresponding to the remainder of the domain is juxtaposed or overlapping. Analysis of erythroblasts derived from mice harboring either a deletion of the two major alpha-globin enhancers or a 16 kb deletion encompassing the genes indicates that the self-interacting domain is still formed in both cases. The results are discussed in the context of a model for chromatin opening within this locus involving recruitment of cohesin complexes by control elements followed by progressive "loop extrusion" until these complexes reach appropriately arranged CTCF binding sites.

This is a highly interesting study that extends previous observations of higher-order chromatin/nuclear structure at the murine alpha-globin locus. The main conclusions would appear to be (1) that formation of the "self-interacting domain" at this region precedes and is independent of globin gene activation, although certain features are accentuated by increased transcription levels; (2) that the flanking regions are not only distinct from the domain, but appear to converge during erythroid

differentiation. The manuscript also presents a novel variation on FISH (RASER) that allows for visualization without DNA denaturation. All of this is likely to be of general interest, and in the context of the "self-interacting domain," represents work that merits a wide audience. Unqualified support for the "loop extrusion" model, however, seems premature. The sequence determinants (if any) for formation of the self-interacting domain remain unclear, and alternatives are not considered here (see below).

--Depictions of the genomic region being investigated span a range of more than 500 kbp. This is useful, at least initially, in demonstrating the specificity of interactions within the alpha-globin domain, the signal-to-noise ratio obtained using the NG Capture-C assay, etc. It does make inspection of the results somewhat difficult, however, and so I think many of the figures would benefit from a narrower range.

--Fig. 1D is described as showing that one of the flanking CTCF/cohesin sites (-39.5) interacts solely with convergent CTCF/cohesin sites at the opposite side of the domain. The data, on the other hand, appears to show a plateau of interactions through the genes and points further downstream, with peaks at sites of CTCF binding. This begs the question: what is the threshold for calling a point on one of these graphs an "interaction"?

--Why not perform the volume analysis for the COMP probe in the AMKO background? Is there still a significant difference between the volume occupied by the COMP probe vs. the flanking BACs in the absence of alpha-globin transcription?

--While alpha-globin transcription as a basis for formation of the self-interacting domain is excluded by the results with the AMKO mice, what about non-genic transcription? The progressive changes observed during differentiation could correspond to increases in overall transcription within the locus, both genic and otherwise. What, for example, might happen if NG Capture-C was performed on cells incubated in the presence of DRB to inhibit transcriptional elongation?

--To what extent can the authors exclude the possibility that the distinction between the self-interacting alpha-globin domain and the flanking regions reflects not the specific activation mechanisms operating within the domain, but rather repressive mechanisms that occur in the flanking regions upon differentiation?

Brown et al. NCOMMS-17-31917-T
Responses to reviewers

Thank you to all the reviewers for their helpful comments, which have guided us towards an improved version and presentation of our work. We have also used this interval to reformat the manuscript according to Nat Comms guidelines, with an Introduction section. The line numbers have therefore altered substantially. We have split Fig. 1 into two figures to allow a fuller description of the two timepoint MFL populations and Fig. 3 into two figures to allow for inclusion of oligo FISH STED data. We also moved Extended Data Fig. 3 to the main body of the manuscript. Additional data as described below have also been incorporated.

Reviewer 1

1a. The title does not describe a novel result. In the last few years, it has become clear that facultative regions are generally open before transcription is maximal (e.g. Therizols et al., 2014, Science 346:1238; Hug et al., 2017, Cell 169:216). The title should state that this paper focuses on correlating imaging and omics data to study the local structure of the mouse α -globin locus.

We agree with the reviewer that there are several lines of evidence that chromatin can be 'open' in the absence of transcription. Therizols et al¹ showed that artificial decondensation without transcription is sufficient to alter positioning of tagged genes from the periphery to the centre of the nucleus. Hug et al² found by Hi-C that TAD boundaries are established in *Drosophila* at the onset of zygotic genome activation, independently of transcription and we did refer to this work in the Discussion. These findings are consistent with the concept that reorganisation of chromatin can proceed in the absence of transcriptional processes, however neither of these studies, nor any others that we know of, have looked at the same level of resolution at an individual mammalian domain of this scale.

NG Capture-C provides high-resolution information on chromatin proximity to a specific viewpoint, garnered from millions of cells. FISH on the other hand provides actual distance measurements between pairs of locations in hundreds of cells. We have not attempted in this work to directly match information from the two techniques but rather use data from both approaches to arrive at an understanding of the fine-scale three-dimensional organisation of the α -globin chromatin domain during erythroid differentiation. We are concerned that the suggested title might imply merely a comparison of two technical approaches. We would therefore like to keep the same title if at all possible, to better reflect the data and conclusions of the manuscript.

We have incorporated additional data into Fig. 2, showing that a self-interacting domain is set up at MFL 0h not only at the α -globin genes, but also at the β -globin genes. This provides additional evidence that domains, characterised at high resolution by Capture-C and not present in ES cells, are already established early in erythroid differentiation.

1b. Results should be compared and contrasted to Ulianov et al., 2017 Epigenetics & Chromatin 10:35, which describes local chromatin changes at the α -globin locus during erythrocyte development in chicken.

With more text for discussion we are pleased to be able to address this point and the following text has been added to the Discussion (lines 308-312):

Within a 2.7Mb segment at the chicken α -globin locus it has also been shown that whilst genic and inter-genic transcription is upregulated in the immediate vicinity of the α -globin genes when they are active, increasing Hi-C contacts do not however automatically equate with increasing transcription³.

and (lines 329-331):

Studies in chicken³ report compaction of the sub-TAD encompassing α -globin when this gene is fully active, however this was based on frequency of chromatin interactions alone without parallel FISH data.

2. The large spread of distance measurements does not allow the authors to conclude that the region is dynamically changing shape (lines 153-154). Differences between cells in the population, such as position in cell cycle, or degree of differentiation, would give similar variation in distance measurements.

We have removed the import that there is a progressive change in shape during erythropoiesis to meet the concerns of both reviewers 1 and 2 (please also see our response to Reviewer 2 below). We agree that the cell populations are not tightly synchronous from a differentiation perspective and Fig 1 now describes the two populations in more detail. Of note, we only selected signal pairs where there was no hint of replicated signal and as this locus replicates very early in S phase, then the majority of cells analysed were in G1. We have added this point to Methods (lines 549-551).

3. Several publications have shown that proximity data correlates best with very short distance data, rather than the median distance. See Giorgetti and Heard, 2016, Genome Biol 17:215; Dekker, 2016, Nat Rev Mol Cell Biol 17:741; Fudenberg and Imakaev M, 2017, Nat Methods 14:673. It is unclear whether some discrepancies in the data would be resolved if the data analysis approach took this into account.

We recognise the important issues raised by the papers listed above and in our work plan we followed the six recommendations of Giorgetti and Heard⁴ as closely as possible:

1. Erythroid cells need to be settled on poly-L-lysine coated coverslips prior to fixation so we could not use the same pool of fixed cells, however cell suspensions were taken from the same cultures for the two techniques.
2. Biological replicates were taken from littermates, as were comparisons between normal and knockout mouse embryos. For experiments involving comparison of 0h and 30h, or normal and mutant material, the RASER-FISH hybridisations and detections were processed concurrently. Our figures are

detailed to allow the reader to follow the data provenance by inclusion of a numerical suffix for all cell populations.

3. The Deltavision Elite is an enclosed system, optimised for stability, with a highly accurate stage, Apochromat objectives and multi-bandpass dichroic mirrors. Imaging was performed sequentially for wavelength at each Z section to minimize the effects of drift. Chromatic aberrations were corrected within the pipeline using values obtained from sub-diffraction-sized beads. All deconvolution employed Huygens software as the gold-standard for image restoration. Z step size was assessed over a range and the selected 150nm allowed substantial over-sampling.
4. Although we used a rigid channel registration, cells were selected from the middle of the image fields and the tolerance level was established by labeling the same probe with two colours as described in Methods (mean 53nm).
5. All Capture-C triplicates were indexed, pooled and processed together.
6. We presented Pearson's correlation of pixel intensities to look at spatial proximity in addition to specific distance measurement. We provided the median coefficients in the text, however an example graph from that analysis is shown below.

The reviewer makes an important point about the examination of very short distance data. We have calculated the 'absolute contact probabilities'⁵ for our hybridisations by using the tolerance data (Fig. S9c)(mean + 3SD = 0.13088) to assess what proportion of our distance measurements could be viewed as co-localised signal. Below we provide a plot of contact probabilities for the BAC data set to match Fig.3c.

At MFL 30h, 25-32% of FISH signals for the flanking regions could be judged as co-localised, compared to 4% for mES cells. Whilst this is striking data, we did not feel that this extra analysis essentially added to the conclusions of the paper, given space restrictions. We are also concerned that the number of data points is small in some instances. However we are happy to include if judged appropriate.

4. The small, often sub-diffraction changes in distance between elements in local gene structure are best measured using super resolution microscopy. This paper would be quite powerful if FISH results were analyzed using STED. The STED panel presented in Figure 3 is provocative, but results using oligo FISH probes sampling the whole locus should be presented. This would go a long way toward substantiating the model and data in Extended Figure 5.

Since our hybridisations gave sharply defined points of fluorescence in three-dimensional space where a centroid to the group of pixels could be accurately assigned, we felt that analysis on the Deltavision system provided high resolution measurements, in line with the recommendations by Giorgetti et al⁶. Performing three-colour STED is not a high throughput approach to allow us to gather the number of measurements required, although we recognise the value of validating our model by this approach. Therefore as requested we present additional scores and images using oligo-FISH probes (Fig. 5c). In these hybridisations we were also able to measure the angle in 3D space occupied by COMP in relation to the edges of the domain (Fig 5d).

5. The loop extrusion model should be presented as speculative.

We note the comments on this issue from both Reviewer 1 and 3 and have amended

the Discussion accordingly (lines 287-298).

6. There is no evidence presented that any cis elements (including any enhancers or promoters) are involved with local structure changes. This conclusion should be removed from the discussion.

The reviewer is correct that we do not present positive evidence of a requirement for specific cis elements in domain formation. Our discussion of potential roles for MCS-R3 and MCS-Rm was offered as a discussion point only, and not a conclusion. We have amended the Discussion to clarify this (lines 340-345).

Specific comments:

1. The transcriptional analysis could be clarified and expanded. A) Another repeat of the α -globin RT-qPCR would help - the error bars in Figure 1b are so large that it appears there is no difference in transcription between 6 and 24 hours. B) What happens with transcription outside of the α -globin locus?

1A) We have expanded the figure describing the cell populations to additionally include Ter119, cytopins and RNA-FISH for nascent transcripts. We recognize that biological replicates of cultured primary cells can vary, therefore, since all comparisons throughout the manuscript are solely made between MFL 0h and 30h, we have restricted the RTqPCR data to those two time points, in support of all the other characterisation of the two populations now presented in the revised manuscript.

1B) We have unpublished RNA-seq data comparing transcription from genes in the region at MFL 0h and 24h, which is part of a large study by a graduate student that cannot be included in this paper. However some of those data are useful here in response the reviewer's question and below we show RNA-seq data for genes in the immediate vicinity of α -globin. The log₂-fold change column indicates that all those surrounding genes are upregulated, in line with the finding of Ulianov et al³.

[Editorial Note: Unpublished data has been redacted from the Peer Review File by the Editorial Team as per Author Request.]

For a broader look at a 6MB surrounding region, please see our response to Reviewer 3.

2. TER119 levels should be shown at 0 and 30 hrs and cytopsin results presented to help authenticate the in situ differentiation. The manuscript text should be reviewed and references to 30 hr cells as intermediate erythroblasts should be deleted.

In Fig. 1 we now include Ter119 FACS panels with cytopsin examples and RNA-FISH scores at 0h and 30h to better define the two populations. We have removed references to MFL 30h cells as intermediate erythroblasts as requested.

3. In the Image Analysis section under Methods, it is stated that signal centers were chosen by hand. It is more accurate to identify the center of maximum intensity mathematically, rather than by eye. Additionally, many BAC and oligo probes do not give spherical signal, rather it can be elongated or lobated. Centroid identification needs to take this into account.

We apologise that we have not described our analysis pipeline with sufficient clarity. The rough locations of the signal pairs are identified with a mouse click but after that the centroid of each signal is calculated mathematically within the analysis algorithm. We have amended the Methods text (lines 551-561) to clarify this point. Centroid identification within the algorithm does indeed take into account the shape of the segmented foci, rather than using a spherical parameterisation. The full algorithm will be available for inspection on acceptance (https://github.com/dwaithe/foci_measurements).

4. Only one CTCF site was used as an anchor for the 3C experiments. The downstream flanking site should also be used. Why is there no peak at this second site showing interaction with the CTCF-39.5 in Figure 3a?

We have now performed Capture-C from the downstream CTCF sites viewpoint as requested and this is shown in a new Fig. 1 where we define the cell populations used, the FISH probe sets and the basic features of the domain. We also show a track from this viewpoint in our analysis of the DKO mouse (Fig. 7).

As we showed in Hanssen et al⁷ and in these new data, the interaction profiles from any of the CTCF-bound boundary elements do not show a punctate interaction pattern with other CTCF sites, rather we see a broad domain of interactions with the opposite flank which is tissue specific. Moreover in Hanssen et al, we show the same pattern of interactions when capturing from proximal promoter elements outside of the globin self interacting domain (see Figure 2 Hanssen et al). This suggests that the formation of the α -globin self-interacting domain has the effect of bringing the flanking regions together in a general manner (as depicted in Fig. S5) rather than specifically tethering a CTCF-bound region to another CTCF-bound region, although this effect is mediated by the boundary action of these sites (see Figure 4 Hanssen et al). We have now clarified these points in the text (lines 277-287). Please also see response to Reviewer 3 on this topic.

The new Capture-C data for the DKO mouse from boundary CTCF viewpoints provides complementary information to the FISH data. Importantly this allows us to conclude that whilst interactions within the domain may be compromised, the gross loop structure of the domain with boundary CTCF interactions is established as for WT, even when α -globin transcription is severely compromised. We have added these findings to Results and Discussion (lines 223-240 and 333-337).

Reviewer 2

The Raser FISH, high resolution microscopy analysis and the NG capture-C data are all performed with high technical rigor in well-designed experiments, and the data is convincing and elegant. The conclusion that domain looping and decompaction is upstream of gene transcription is especially interesting.

We thank the reviewer for this assessment of our work.

My only reservation relates to the dynamics described for the chromatin reconfiguration events. Throughout the manuscript, the authors repeatedly suggest a gradual process, where the activated configuration of the alpha globin domain is found with increasing 'frequency' in later stages of development. This description would only hold if the authors had performed their experiments with MFLs at well delineated developmental stages. The authors state that MFL 0h do not undergo alpha globin transcription, whereas MFL 30h are at peak alpha globin transcription; neither is likely to be entirely correct. Both MFL 0h and MFL 30h are likely to be mixtures of non-transcribing and transcribing cells, albeit at different proportions. (Indeed, the single cell inter-centroid distance data has a much larger range in MFL 30h than in ES cell, consistent with MFL 30h being a mixture of two populations, see Extended Data Figure 2). For MFL 0h, the authors select CD44high Ter119neg cells, which are likely to be a mixture of CD71high and CD71med/low cells; the former would most likely already contain the active chromatin conformation (see Pop et al., PLoS Biology 2010) and are likely to be undergoing alpha globin transcription, though the transcript may not yet reach the threshold required for detection. Similarly, MFL 30h might contain some residual cells that have not yet differentiated. It is therefore not possible to make any statements regarding the dynamics of the chromatin conformational change with this specific dataset. This issue could easily be addressed, though, by re-stating this problem upfront and focusing on the principal findings in this manuscript, which are not affected by this issue.

We tried to be careful with our terminology and did not intend to imply that our 0h and 30h time points represented pure and sharply-defined populations. We have now clarified this point at the beginning of Results (lines 101-112) and in Methods (lines 363-366) and provide additional data defining the populations in the new Fig. 1. Further, we have removed any implication of a gradual dynamic change in conformation.

Reviewer 3

This is a highly interesting study that extends previous observations of higher-order chromatin/nuclear structure at the murine alpha-globin locus. The main conclusions would appear to be (1) that formation of the "self-interacting domain" at this region precedes and is independent of globin gene activation, although certain features are accentuated by increased transcription levels; (2) that the flanking regions are not only distinct from the domain, but appear to converge during erythroid differentiation.

The manuscript also presents a novel variation on FISH (RASER) that allows for visualization without DNA denaturation. All of this is likely to be of general interest, and in the context of the “self-interacting domain,” represents work that merits a wide audience.

We thank the reviewer for this assessment of our work.

--Unqualified support for the “loop extrusion” model, however, seems premature.

We note the comments on this issue from both reviewer 1 and 3 and have amended the Discussion accordingly (lines 287-298).

--The sequence determinants (if any) for formation of the self-interacting domain remain unclear, and alternatives are not considered here (see below).

The reviewer is correct that we have not pinned down precise determinants and as noted by reviewer 2, this is being examined as future work.

--Depictions of the genomic region being investigated span a range of more than 500 kbp. This is useful, at least initially, in demonstrating the specificity of interactions within the alpha-globin domain, the signal-to-noise ratio obtained using the NG Capture-C assay, etc. It does make inspection of the results somewhat difficult, however, and so I think many of the figures would benefit from a narrower range.

We take note that a narrower range could be helpful and have amended figures for clarity. We have left a 500 kb range for the introductory Fig. 1 and also for Fig. 3 where the tracks for control probes extend the range required.

--Fig. 1D is described as showing that one of the flanking CTCF/cohesin sites (-39.5) interacts solely with convergent CTCF/cohesin sites at the opposite side of the domain. The data, on the other hand, appears to show a plateau of interactions through the genes and points further downstream, with peaks at sites of CTCF binding. This begs the question: what is the threshold for calling a point on one of these graphs an “interaction”?

As described in Davies et al⁸, Oudelaar et al⁹ and Hanssen et al⁷, due to the highly quantitative nature of NG Capture-C, data can be analysed without the necessity of distance normalisation or thresholding. The experiments as described represent triplicates of the interactions in erythroid cells (in which alpha globin is active) as compared to triplicates of interactions in mES cells (in which alpha globin is inactive). Using this experimental set up we can statistically determine which individual restriction fragments increase or decrease their levels of interactions with the viewpoint, depending on the activity of the locus. With this approach, we do indeed show this whole plateau of interactions is formed specifically in erythroid cells, rather than forming punctate interactions with specific CTCF bound sites. Please see Hanssen et al for a full description and our response to Reviewer 1 above. The reviewer is correct that we did not clearly discuss the plateau in the original text, which we have now amended (lines 277-287).

--Why not perform the volume analysis for the COMP probe in the AMKO background? Is there still a significant difference between the volume occupied by the COMP probe vs. the flanking BACs in the absence of alpha-globin transcription?

This is a great idea and something we considered. Unfortunately the size of the deletion in this mutant means a substantial reduction in the signal volume of the COMP BAC to a level where we would not be confident of the consistency of our measurements.

--While alpha-globin transcription as a basis for formation of the self-interacting domain is excluded by the results with the AMKO mice, what about non-genic transcription? The progressive changes observed during differentiation could correspond to increases in overall transcription within the locus, both genic and otherwise. What, for example, might happen if NG Capture-C was performed on cells incubated in the presence of DRB to inhibit transcriptional elongation?

This is an important question. We have performed an experiment as suggested by the reviewer where we globally inhibit transcription in differentiating erythroblasts at MFL 30h by a 3h exposure to DRB. The results are presented in an additional Fig. 9 showing that despite successful knockdown of transcription (Fig. S7), the domain structure is unaltered. This indicates that overall transcription within the domain is not contributing to maintenance of the domain structure.

--To what extent can the authors exclude the possibility that the distinction between the self-interacting alpha-globin domain and the flanking regions reflects not the specific activation mechanisms operating within the domain, but rather repressive mechanisms that occur in the flanking regions upon differentiation?

This is an interesting proposal. RNA-seq data across a 6Mb region encompassing the α -globin domain taken at MFL 0h and 24h however does not support this concept. Within this region but excluding the α -globin domain itself, there are 35 genes with quantifiable expression levels. Of these, 14 genes showed no significant change in expression level but of the of 21 genes that do show a significant change, there is a general tendency towards upregulation (15 genes) rather than repression (6 genes). Please also see specific comment response 1B to Reviewer 1.

References

- 1 Therizols, P. *et al.* Chromatin decondensation is sufficient to alter nuclear organization in embryonic stem cells. *Science* **346**, 1238-1242, (2014).
- 2 Hug, C. B., Grimaldi, A. G., Kruse, K. & Vaquerizas, J. M. Chromatin Architecture Emerges during Zygotic Genome Activation Independent of Transcription. *Cell* **169**, 216-228 e219, (2017).
- 3 Ulianov, S. V. *et al.* Activation of the alpha-globin gene expression correlates with dramatic upregulation of nearby non-globin genes and

- changes in local and large-scale chromatin spatial structure. *Epigenetics & Chromatin* **10**, 35, (2017).
- 4 Giorgetti, L. & Heard, E. Closing the loop: 3C versus DNA FISH. *Genome Biol* **17**, 215, (2016).
- 5 Cattoni, D. I. *et al.* Single-cell absolute contact probability detection reveals chromosomes are organized by multiple low-frequency yet specific interactions. *Nat Commun* **8**, 1753, (2017).
- 6 Giorgetti, L., Piolot, T. & Heard, E. High-Resolution 3D DNA FISH Using Plasmid Probes and Computational Correction of Optical Aberrations to Study Chromatin Structure at the Sub-megabase Scale. *Nuclear Bodies and Noncoding Rnas: Methods and Protocols* **1262**, 37-53, (2015).
- 7 Hanssen, L. L. P. *et al.* Tissue-specific CTCF-cohesin-mediated chromatin architecture delimits enhancer interactions and function in vivo. *Nat Cell Biol* **19**, 952-961, (2017).
- 8 Davies, J. O. *et al.* Multiplexed analysis of chromosome conformation at vastly improved sensitivity. *Nat Methods* **13**, 74-80, (2016).
- 9 Oudelaar, A. M., Davies, J. O. J., Downes, D. J., Higgs, D. R. & Hughes, J. R. Robust detection of chromosomal interactions from small numbers of cells using low-input Capture-C. *Nucleic Acids Res* **45**, (2017).

REVIEWERS' COMMENTS:

Reviewer #1 (Remarks to the Author):

The authors have performed a number of experiments to address the reviewer's comments. We are satisfied with the new data. The revised manuscript will make a very nice contribution to the field.

Reviewer #2 (Remarks to the Author):

The authors have responded to all my concerns and have further strengthened this interesting work.

Reviewer #3 (Remarks to the Author):

In response to reviewer comments, the authors have made a number of modifications and edits to the text, and to selected figures as well. The major new experiment included in this version is NG Capture-C performed in cells treated for 3 hours with the general transcription inhibitor DRB. Altogether these changes, in my view, improve the manuscript and successfully address the concerns that I had with the original submission. As it stands, I find the work reasonably convincing, and I believe there will be a general interest in the major findings regarding changes in locus-wide conformation in the nucleus that occur at an active gene locus prior to gene activation.

Regarding the DRB experiment, however, there is still a minor concern with its presentation in the text. DRB is an inhibitor of transcriptional elongation, and generally has little effect on initiation. Thus, it is still not possible to claim that "overall transcription within the domain is not contributing to maintenance of domain structure," given that RNA pol II is likely still recruited to selected sequences within the locus, where it initiates and proceeds for a few dozen bases or so before stalling. A minor tweak/clarification to the text is probably warranted, if only to make various journal clubs sitting through a presentation of this study work a little harder for their criticisms.

Brown et al. NCOMMS-17-31917-T
Responses to reviewers – 2nd revision

Reviewers 1 and 2

We are pleased that the reviewers ask for no further changes.

Reviewer 3

Regarding the DRB experiment, however, there is still a minor concern with its presentation in the text. DRB is an inhibitor of transcriptional elongation, and generally has little effect on initiation. Thus, it is still not possible to claim that “overall transcription within the domain is not contributing to maintenance of domain structure,” given that RNA pol II is likely still recruited to selected sequences within the locus, where it initiates and proceeds for a few dozen bases or so before stalling. A minor tweak/clarification to the text is probably warranted, if only to make various journal clubs sitting through a presentation of this study work a little harder for their criticisms.

Reviewer 3 requests that we clarify the effect of DRB as limited to transcriptional elongation. We have now made this clear in the Abstract, Introduction, Results, Discussion and legend to Figure 9, as shown in tracked changes.